# MULTITASK REINFORCEMENT LEARNING BY OPTIMIZING NEURAL PATHWAYS

## ABSTRACT

Reinforcement learning (RL) algorithms have achieved great success in learning specific tasks, as evidenced by examples such as AlphaGo or fusion control. However, it is still difficult for an RL agent to learn how to solve multiple tasks. In this paper, we propose a novel multitask learning framework, in which multiple specialized pathways through a single network are trained simultaneously, with each pathway focusing on a single task. We show that this approach achieves competitive performance with existing multitask RL methods, while using only 5% of the number of neurons per task. We demonstrate empirically the success of our approach on several continuous control tasks, in both online and offline training.

## 1 INTRODUCTION

Our brain processes different languages, helps us perceive and act in a 3D world, coordinates between organs, and performs many other tasks. The brain continuously learns new things without catastrophic forgetting due to its plasticity (Zilles, 1992; Drubach, 2000; Sakai, 2020; Rorke, 1985), i.e., its ability to continually strengthen more frequently used synaptic connections and eliminate synaptic connections that are rarely used, a phenomenon called *synaptic pruning* (Feinberg, 1982). In this way, the brain creates *neural pathways* to transmit information. Different neural pathways (Rudebeck et al., 2006; Paus et al., 1999; Jürgens, 2002; Goodale et al., 1994) are used to complete different tasks. For example, the visual mechanisms that create the perception of objects are functionally and neurally distinct from those controlling the pre-shaping of the hand during grasping movements directed at those objects (Goodale et al., 1994). In deep learning architectures, the hope is that gradient descent, possibly augmented by attention mechanisms, will create these types of pathways automatically. However, when performing multi-task learning, this approach can be insufficient to prevent interference between tasks, and often leads to loss of plasticity. This problem is especially exacerbated in reinforcement learning, where different tasks require policies which generate different data distributions.

In this paper, we take inspiration from the presence of distinct neural pathways for different tasks in natural brains, and propose a novel approach to implement it in deep reinforcement learning. Similarly to synaptic pruning, we aim to identify the important connections among the neurons in a deep neural network that allow accomplishing a specific task. However, we leverage insights from recent *lottery ticket hypothesis* (Frankle & Carbin, 2019; Lee et al., 2018; Tanaka et al., 2020; Wang et al., 2020a) literature to construct *task-specific neural pathways* which utilise only a very small fraction (5%) of the parameters of the entire neural network, yet demonstrate expert-level performance in multitask reinforcement learning in both *online* and *offline* settings. The neural pathways are allowed to overlap, which enables leveraging information from multiple tasks in the updates of the same parameters, in order to improve generalization.

**Contributions:** We propose *Neural Pathway Framework* (NPF), a novel multitask learning approach that generates neural pathways through a large network that are specific to single tasks. Our approach can be easily integrated with any online or offline algorithm without changing the learning objective[1]. Because the shift in data distribution during online reinforcement learning training makes it challenging to find neural pathways by using lottery network search techniques (Frankle & Carbin, 2019; Alizadeh, 2019; Wang et al., 2020a), we propose a *data adaptive pathway discovery* method for

---

[1]Code: https://github.com/anomICLR2023/2904

the online setting. We demonstrate our approach in both offline and online multitask reinforcement learning, using Mujoco-based and MetaWorld environments.

Our method is frugal in terms of the number of neurons, and multiple tasks are trained in parallel, thus saving significant training time. The algorithm finds neural pathways which use only $5\%$ of the neural network weights for each task. Since our method requires a small fraction of weights to complete a task, it can provide fast inference (Molchanov et al., 2017; Luo et al., 2017), reducing energy consumption (Yang et al., 2016). Thus, our method is especially suitable for real-world applications in which large datasets need to be processed in real-time (e.g., self-driving cars, deploying bots in games, financial data analysis) or which are deployed on low-resource devices (i.e., embedded systems, edge devices, etc.).

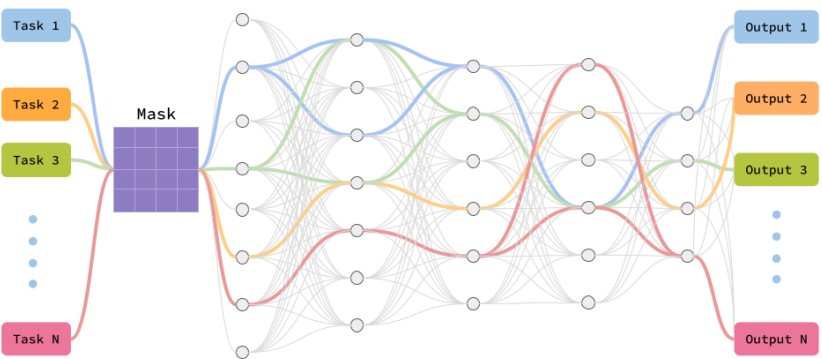

Figure 1: *During evaluation* of a trained agent (global model), for any given task our proposed method activates specific part of the neural network.

## 2    RELATED WORK

Many recent works point out that training a deep neural network for *more than one task* becomes difficult due to *gradient interference*, i.e., gradients for different tasks pointing in very different directions. Recent work proposes several possible solutions, such as constraining the conflicting gradient update (Yu et al., 2020; Suteu & Guo, 2019; Du et al., 2020; Chen et al., 2017; Sener & Koltun, 2018), constraining data sharing among irrelevant tasks (Yu et al., 2021; Fifty et al., 2021), and learning the underlying context of relevant tasks (Sodhani et al., 2021). In all of these cases, success depends on the assumption that there is an underlying shared structure among these tasks, and on how well this structure can be captured during training. Many multitask learning methods propose complex ways to modularize the neural network, which allow sharing and reusing network components across tasks (Rusu et al., 2016; Fernando et al., 2017; Rosenbaum et al., 2017; Devin et al., 2016; Misra et al., 2016; Yang et al., 2020). In this paper, we propose a completely new way of tackling multitask problems, and we show that a single deep neural network is sufficient for learning multiple complex tasks, by having multiple pathways through the network.

Recent advances in finding *sparse networks* have proven that there exist *sub-networks* which contain a small fraction of the parameters of the dense deep neural network yet retain the same performance. There is a range of techniques to find such sub-networks through an *iterative update* during training (Frankle & Carbin, 2019; Feinberg, 1982; Gale et al., 2019; Blalock et al., 2020; Ohib et al., 2019; Yang et al., 2019). Simpler alternatives find a sub-network *single-shot* (Lee et al., 2018; Wang et al., 2020b) at weight initialization. Building upon a three-decade-old saliency criterion used for pruning trained models (Mozer & Smolensky, 1988), a recent technique to prune models at initialization was proposed by Lee et al. (2018). Utilizing this criterion, they are able to predict, at initialization, the importance that each weight will have later in the training process. However, premature gradient-based pruning at initialization can lead to layer collapse: the premature pruning of an entire layer can render a network un-trainable. Many techniques (Wang et al., 2020b; Feinberg, 1982) try to maximize useful gradient flow through the deep layers. In order to discover *neural pathways*, we can leverage these methods in order to find *task-dependent sub-networks*, as long as we can avoid layer collapse.

Figure 2: Example of learning two tasks in the *offline setting*. *During training*, the global model (purple) communicates with the task specific local models (green). Each local model computes the gradient of the *task specific weights* and sends them to the shared global model (pink arrow). Network weights are updated in the global model and then shared (blue arrow) with the local models at every training step.

## 3 BACKGROUND

**Multitask Reinforcement Learning:** We consider learning in a Markov Decision Process (MDP) described by the tuple $(S, A, P, R)$ consisting of states $s \in S$, actions $a \in A$, transition dynamics $P(s'|s, a)$, and reward function $R(s, a)$. We use $s_t$, $a_t$ and $r_t = R(s_t, a_t)$ to denote the state, action and reward at time step $t$, respectively. A trajectory consists of a sequence of states $s_t$, actions $a_t$ and rewards $r_t$: $\tau = (s_0, a_0, r_0, s_1, a_1, r_1, ..., s_T, a_T, r_T)$. For continuous control tasks, we consider an infinite horizon, $T = \infty$ and the goal is to learn a policy which maximizes the expected discounted return $\mathbb{E}[\sum_{t=0}^{T} \gamma^t r_t]$. In the multitask setup, we consider a finite set of $N$ tasks with respective reward functions $\{R_n\}_{n=1}^{N}$ and optimal policies $\{\pi_n^*(a|s)\}_{n=1}^{N}$.

**Offline Reinforcement Learning:** In offline reinforcement learning, instead of obtaining data through environment interactions, we have access to a previously collected, fixed dataset $D = \{(s_i, a_i, s_i', r_i)\}_{i=1}^{L}$ with $L$ transitions collected using a behavior policy $\pi_\beta(a|s)$. For multitask offline learning, we have $N$ datasets, $\{D_n\}_{n=1}^{N}$ collected for each task. Offline training does not perform well when trained naivelt with batch data, due to extrapolation errors caused by data distribution mismatch between the samples collected from behavior policy $\pi_\beta$ and the current policy $\pi_\theta$. In this work, we use two standard offline RL algorithms, *Batch-Constrained deep Q-learning* (BCQ, Fujimoto et al. (2018)) and *Implicit Q-learning* (IQL, Kostrikov et al. (2021)) to build our offline experiments.

**Online Reinforcement Learning:** In the online RL setting,. the agent is expected to interact with the environment and leverage the collected experience to learn how to maximize its expected return. Neural pathways can be integrated into any online RL algorithm and trained for multitask objectives. To demonstrate the effectiveness of our method in the online setting, we use *Soft-Actor-Critic* (SAC) (Haarnoja et al., 2018) in our experiments. SAC optimizes an entropy-regularized objective, in order to drive the agent to more exploratory behavior.

## 4 PROPOSED APPROACH

We now introduce the *Neural Pathway Framework (NPF)* and its practical implementation for multitask learning in two steps: (1) finding the neural pathway for each task and (2) training the neural pathways for multiple tasks simultaneously. To discover task-specific neural pathways, we build on the idea of finding a *lottery-ticket sub-network* (Frankle & Carbin, 2019) (i.e., a sub-network that provides similar performance

Table 1: Comparison of pruning methods against the neural-pathway-framework (NPF).

| Important Features | Pruning Methods | NPF |
|---|---|---|
| Finding sparse sub-network | ✔ | ✔ |
| Data dependency | ✔/✗ | ✔ |
| Tackles data-distribution shift | ✗ | ✔ |
| Multi-task learning | ✗ | ✔ |

to the original network) using existing pruning methods, as detailed in Sec.4.1. We then describe how to use NPF in both offline (Sec. 4.2) and online (Sec. 4.3) RL settings.

### 4.1 DISCOVERING NEURAL PATHWAYS

Similarly to the *Lottery ticket hypothesis* (Frankle & Carbin, 2019) in supervised learning, we assume that a dense network contains a sub-network that can be initialized such that, when trained in isolation, it can match the performance of the original network. We call this sub-network *a Neural Pathway*. In the context of multitask training, a deep network will have multiple neural pathways, one for each task, which can share neurons.

Formally, we consider a feed-forward network $f(x, \theta)$ with initial parameters $\theta_0 \sim D_\theta$. A Neural Pathway activates a sub-network $f(x, \theta \odot m)$ by using a mask $m \in \{0, 1\}^{|\theta|}$, where $m$ is fixed during training. From the lottery ticket hypothesis, there exists $m$ which leads to better performance, less training time, and $\|m\|_0 \ll |\theta|$ (fewer trainable parameters). For multi-task learning, as shown in Figure 1, multiple neural pathways are activated with $n$ masks $\{m_1, m_2, ..m_n\}$, one for each task.

In order to learn the neural pathways, we build on *Single-shot Network Pruning* (SNIP) (Lee et al., 2018; Arnob et al., 2021b), a pruning technique which has proven to perform well in a single-task setting, leading to up to $95\%$ sparse network. SNIP identifies important connections by using a sensitivity measure, defined as the influence of each weight on the loss function. Formally, the effect of weight $\theta_q$ on the loss is: $S(\theta_q) = \lim_{\epsilon \to 0} \left| \frac{\mathcal{L}(\theta_0) - \mathcal{L}(\theta_0 + \epsilon \delta_q)}{\epsilon} \right| = \left| \theta_q \frac{\partial \mathcal{L}}{\partial \theta_q} \right|$, where $\delta_q$ is a vector whose $q_{th}$ element equals $\theta_q$ and all other elements are 0.

We note that, while we build our approach by leveraging pruning methods, our goal is not to prune weights, but to flexibly use the capacity of the network to learn many tasks. Therefore, our approach needs to be data-dependent. Moreover, in the online RL case, NPF needs to be robust to shifts in the data distribution due to a changing policy. Thus, for $n^{th}$ task, we use the following objective:

$$\mathrm{m}_n = S(\theta_q; D_n) = \lim_{\epsilon \to 0} \left| \frac{\mathcal{L}(\theta_0; D_n) - \mathcal{L}(\theta_0 + \epsilon \delta_q; D_n)}{\epsilon} \right| = \left| \theta_q \frac{\partial \mathcal{L}}{\partial \theta_q} \right|. \tag{1}$$

Table 1 summarizes the differences between NPF and pruning methods.

## 4.2 OFFLINE MULTITASK LEARNING WITH NPF

**Training a Neural Pathway:** First, we identify the top $5\%$ most important weights for each task independently using Equation (1), which will be remembered using binary masks, one for each task, in which 1 marks trainable weights. We never set any weights to zero. But, due to masking, only the subset of the weights contained in the neural pathway participate in task-specific inference and updates. Thus, *for any given task, the mask $m_n$ activates a pathway of neurons for forward propagation and backward propagation.*

**Simultaneous Multitask Training:** In order to train multiple tasks using offline RL, we must have corresponding training datasets for all the tasks $\{D_n\}_{n=1}^N$. We run simultaneous parallel training processes, one for each task, which use only the task-specific expert data. A local model $L(\theta * m_n)$ is initialized for task $n$ with the shared-model/global-model $L(\theta)$ parameters. As shown in Figure 2, the task-specific mask $m_n$ is applied to get the local model $L(\theta * m_n)$.

For each task $n$, at each training step, we synchronize the local $L(\theta * m_n)$ model weights with the global model and sample a random batch of data from dataset $D_n$. The local model computes the gradient $\nabla_{\theta * m_n} J(D_n)$ of the objective function and sends the gradients to the global model (see Fig. 2). Due to masking, gradients of the weights that are irrelevant for the task appear to have the value 0 to the local-model at all times. We run all the process on a single GPU and compute synchronous parameter updates (Niu et al., 2011; Mnih et al., 2016). The pseudocode is given in Algorithm 1.

Offline RL algorithms utilise multiple deep neural networks with different objective functions. For example, IQL has an actor network $\pi(\theta)$, a critic network $Q(\phi)$ and a value network $V(\psi)$, each having their own objective function (see Appendix A.1). Thus for the $n$th task, we use the scoring function 1 to find masks $(m_n^\theta, m_n^\phi, m_n^\psi)$ for each network. The global model $\pi(\theta), Q(\phi), V(\psi)$ is shared with all $N$ processes, where each process is trained to optimize for a single task using the local models $\pi(\theta * m_n^\theta), Q(\phi * m_n^\phi), V(\psi * m_n^\psi)$. The global model stores all the trained pathways. As shown in Figure 1, during evaluation we use the task-specific mask to activate each task-specific pathway.

## 4.3 ONLINE MULTITASK LEARNING WITH NPF

In this section, we discuss the compatibility of NPF with online RL. We empirically show that *due to distribution shift in online training, single-shot sub-network discovery techniques fail to maintain good performance. We propose a novel approach to address the data distribution shift and show its ability to perform multitasking using neural pathways.*

The lottery ticket hypothesis work in supervised learning (Lee et al., 2018; Wang et al., 2020b; Tanaka et al., 2020) and offline RL (Arnob et al., 2021b) only considers the case of static datasets, without

---

**Algorithm 1** Multitask Offline Training

---

# Find the Task Specific Weight Masks:

▷ $m_1, m_2 .. m_N$ = FindNeuralPathways $\left(D_1, D_2, ...D_N\right)$; using Equation 1.

#Initialize a model for each task and consider synchronous gradient update:

▷ Initialize local-model, global-model

#Training loop:

**for** Training steps **do**

   #sync-weight

   ▷ local-model = mask$\left($global-model$\right)$ # $L(\theta * m_n)$; mask the $n^{th}$ task-specific weights

   ▷ Sample $(s, a, s') \sim D_n$

   ▷ local-model.loss()

   ▷ local-model.backward() # $\nabla_{\theta \odot m_n} J(D_n)$ ; compute gradient of masked weights

   ▷ sync-gradient$\left($local-model, global-model$\right)$

   ▷ global-model.optimizer.step()

**end for**

#Evaluation loop:

▷ local-model = mask$\left($global-model$\right)$ # sync-weight

▷ Evaluate( local-model)

---

focusing on the effects of out-of-distribution data shifts. However, in online RL, the agent's policy is always changing, which leads to significant changes in the data distribution.

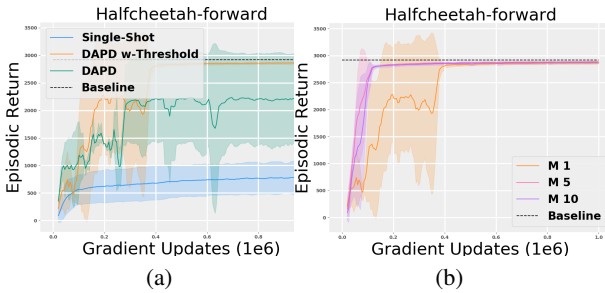

Figure 3: Performance comparison of (a) Single-Shot and *Data Adaptive Pathway Discovery* (DAPD) (with and without Threshold Stopping Criterion) on HalfCheetah run forward,(b) DAPD with Different Moving Average (M=1, 5, 10) on HalfCheetah run forward.

To illustrate the impact of nonstationaritym we train *Soft Actor Critic* (SAC, Haarnoja et al. (2018)) using NPF on the HalfCheetah forward-run task and compare its performance with the final performance of the *fully dense SAC baseline*. Unlike in offline setting, we do not have access to a prior collected dataset. Thus, to configure neural-pathway, we collect samples using a randomly parameterized policy and use those samples to find a sub-network which optimizes Equation (1). The learning curve in Fig. 3 plots the episodic return over 1M gradient updates. We see a large gap in performance between the baseline and *single-shot pathway discovery* (blue). Four more experiments, discussed in the Appendix (Figures 13), display similar results.

We hypothesize the observed performance drop is due to the fact that the changes in behavior policy cause a data distribution shift, and the pathway discovery depends strongly on the data. To tackle this problem, we propose a *dynamic iterative pruning* approach, where pathways are continually updated using the most recently collected training samples.

**Data Adaptive Pathway Discovery Under Data Distribution Shift:** To handle changing data distribution in online RL, we propose *Data Adaptive Pathway Discovery* (DAPD), where we continue to re-configure the pathway as the policy improves and the data distribution consequently changes during the training. We use a separate *temporary replay buffer* $TD_n$, one for each task, that *only contains immediate episodic data* collected by the agent. Only the most recent dataset stored in $TD_n$ is used to re-configure the pathway. Once we reach a minimum threshold $T_H$ on the episodic return, we stop this process. In our experiment in Fig 3(a) we use episodic return $T_H = 2000$ as the threshold value – a hyper-parameter that can be easily tuned to stabilize training.

During the initial training phase, we observe high variation in the pathway. Thus, to slow down variations and stabilize to a suitable pathway, we keep re-configuring using the average of the previous $M$ iteration scores (calculated using Eq. (1):

$$\text{m}_n = S_{avg}(\theta_q; TD_n) = \sum_{m=1}^{M} \lim_{\epsilon \to 0} \left| \frac{\mathcal{L}(\theta_0; TD_n) - \mathcal{L}(\theta_0 + \epsilon \delta_q; TD_n)}{\epsilon} \right| = \sum_{m=1}^{M} \left| \theta_q \frac{\partial \mathcal{L}}{\partial \theta_q} \right|. \quad (2)$$

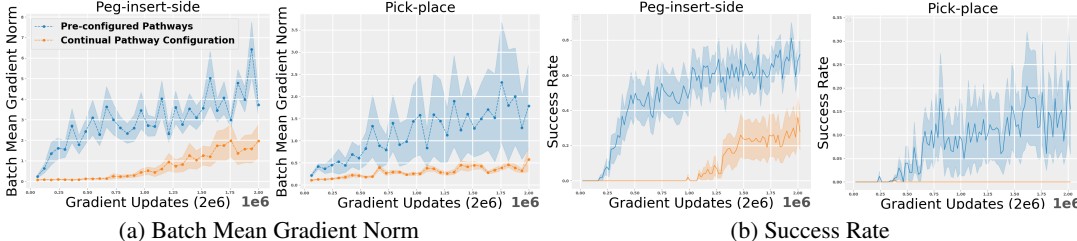

(a) Batch Mean Gradient Norm          (b) Success Rate

Figure 4: When we try to *configure pathways for all the tasks at the same time* (orange), (a) parameters of the harder tasks do not change from the values at initialization due to small changes in gradients. This results in unsuccessful task learning and can be seen in (b). We find an improvement in performance when *pathways are pre-configured separately before multitask training* (blue).

Table 2: Normalized score of offline RL algorithms. We compare NPF performance with the *single task experts* on `HalfCheetah` and `Quadrupod` multitasks.

| Experiment | NPF | |
|---|---|---|
| | BCQ | IQL |
| `HalfCheetah` multitask | $87.20 \pm 2.34$ | $89.67 \pm 12.44$ |
| `HalfCheetah` constrained speed | $100 \pm 0.038$ | $99.88 \pm 0.40$ |
| `Quadrupod` multitask | $115.35 \pm 0.70$ | $116.71 \pm 0.57$ |

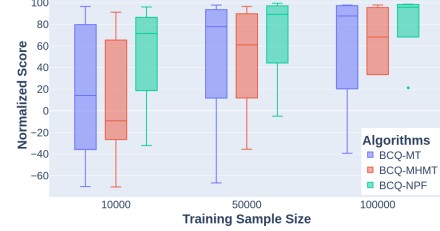

Figure 5: We compare the sample complexity analysis of BCQ+NPF with BCQ-MT (Multitask) and BCQ-MHMT (Multihead Multitask) baselines on `HalfCheetah` multitask.

In Fig. 3(b), we perform a hyper-parameter sweep and observe that $M = 10$ leads to stable learning curves and better performance. Further experiments in the Appendix (Fig. 14) support this claim. Our procedure is summarised in Algorithm 2 (see Appendix) and can be easily integrated with any online RL method.

**Online Multitask Training and Addressing Gradient Interference:** Even though NPF compartmentalizes the parameter training of a neural network very well in offline RL, optimizing pathways for multiple tasks becomes difficult in the online RL setting. Due to significant parameter sharing among tasks, we cannot completely avoid gradient interference. While experimenting with Metaworld (Yu et al., 2019), we found that DAPD optimizes pathways for easier tasks very quickly. In comparison, we see only insignificant changes in gradient updates for the more challenging tasks. A neural pathway is a *task-specific group of parameters* within a network. Thus, to observe changes in the parameters, we compute gradient-norm $\|\frac{dw}{dL}\|$ of *group the parameters* for a batch of samples. See in Figure 4(a), where our initial experiments with parallel learning (*orange*) show little to no gradient-norm change in the task-specific parameters. This implies the parameters have not changed much from their initialization. We find that the success rate (see Figure 4(b)) of the tasks correlates with the gradient change as well.

To overcome radient interference, we run a pretraining loop to learn pathways. We initialize a separate network, one for each task, with same weights and optimized for task-specific pathways for 10K gradient steps. At the end of this pretraining stage, we average the weights of the overlapping parameters and keep the pathways fixed for the rest of the training. We conduct parallel online data-collection and during training, we average the gradients of the shared parameters. For the Metaworld multitask benchmark this method shows a significant improvement in gradient change for the harder tasks (see Figure 4(a) (*blue*)). We discuss the overall performance of our method and compare with baselines in the following section.

## 5 EXPERIMENTS

We evaluate our proposed method on continuous control multi-task experiments in both offline and online RL settings. In offline RL, it is standard to compare performance with an expert (Fujimoto et al., 2018; Fu et al., 2020; Gülçehre et al., 2020). Thus we evaluate the effectiveness of our offline method (Section 5.1) by computing a normalized score with respect to a provided expert. Further we use Meta-World (Yu et al., 2019) multi-task environments to compare with other baselines in both offline (Section 5.1) and online (Section 5.2) settings.

Table 3: Performance comparison in Meta-World offline. We compare the final success-rate (mean and std over 10 seeds) of NPF on MT10 tasks with Offline-MT and Offline-MHMT baselines on offline RL algorithms.

| Experiment | NPF | | Offline MT | | Offline MHMT | |
|---|---|---|---|---|---|---|
| | BCQ | IQL | BCQ | IQL | BCQ | IQL |
| MT-10 tasks | **100 ± 0.0** | **97.3 ± 7.17** | 81.5 ± 24.15 | 79,1 ± 26.81 | 95.9 ± 10.44 | 96.5 ± 7.10 |

## 5.1 OFFLINE MULTI-TASK PERFORMANCE EVALUATION

We train *Soft-Actor-Critic* (SAC) for each task to collect an expert dataset (see Section B.2). We use BCQ (Fujimoto et al., 2018) and IQL (Kostrikov et al., 2021) for our offline training and conduct our experiments on multiple high dimensional continuous control domains. For the offline RL setup, when designing experiments, we focus on two criteria. *First*, how does our method perform compared to the performance of *single task experts* and *second*, how does our method perform when compared to common baseline. When comparing to an expert, we carefully select a range of multitask benchmarks ensuring that our method (1) is tested with a diverse range tasks (Multitask `HalfCheetah`) (2) can yet be performed in a controlled tasks that are similar (`HalfCheetah` Constrained Velocity) in nature and (3) can master skills that can easily be transferred in real world applications (Multitask `Quadrupod`). To compare to common baseline algorithms, we evaluate on the Meta-World Multi-Task benchmark.

**Multitask `HalfCheetah`:** We train `HalfCheetah` for five different kinds of task (Finn et al., 2017; Rakelly et al., 2019) where it needs to (i) run forward, (ii) run backward, (iii) jump, (iv) jump while running forward and (v) jump while running backward. It is important to note that the gait movements of these tasks are very divergent and snapshots are provided in the Appendix.

**Constrained Velocity `HalfCheetah`:** We consider another variation of the `HalfCheetah` environment where it only needs to go forward but we constrain the velocity of the `HalfCheetah` to six different target values from 0.5 to max speed 3.0 (Finn et al., 2017; Rakelly et al., 2019). The success of multitask training at different velocities has greater impact in real-world robotic tasks, where agents are expected to run at different speeds in different kinds of terrain (Lee et al., 2020; Fu et al., 2021; Yang et al., 2021; Hwangbo et al., 2019).

**Multitask `Quadrupod`:** Multitask `Quadrupod` consists of (i) run forward, (ii) run backward, (iii) hopturn: hop and turn to left followed by turning back to initial position and (iv) sidestep: take a step left and take a step to right to the initial position. This simulated environment is commonly used in simulation to real-world transfer experiments (Li et al., 2021; Peng et al., 2020; Yang et al., 2021).

Each task in a multitask experiment is trained with an equal number of gradient updates. We compute the average episodic return over 10 episodes of the trained agent. Our results in Table 2 are reported over seeds 0-4 of the Gym simulator and the network initialization. We run the `Halfcheetah` experiments for 1M and `Quadrupod` for 500k gradient updates. For evaluation, we compare with expert performance and normalize the episodic return using standard offline evaluation metric (Fu et al., 2020), $normalized\ score = \left(\frac{score - random\ score}{expert\ score - random\ score} * 100\right)$, where *random score* is generated by unrolling a randomly initialized policy and averaged over 100 episodes. A score of 100 represents the average returns of a domain-specific expert. *In this paper, for multitasks, we always report the mean and standard-deviation over the normalized scores.* Our proposed method achieves expert-like performance for most of the tasks in 3 sets of experiments. Results are in Table 2. Per task performance is in the Appendix (Table 6).

In Table 2, we find the `Halfcheetah` multitasks to be more challenging for NPF. To further validate the reliability of our method, we conduct a *sample complexity analysis* Arnob et al. (2021a). In Figure 5, we compare NPF performance with baseline performance for different reduced training sample size. We use two baselines: (a) multi-task algorithms (MT): where we train offline algorithms for multiple tasks without changing the objective function or the neural architecture. (b) A multi-head (MHMT) variant where we use independent heads for tasks. Both baselines are non-pruned (use 100% of network weights), while our method uses only 5%. The interquantile plot (Figure 5) of normalized scores shows a consistent improvement when BCQ is trained with NPF.

**Meta-World** (Yu et al., 2019) is a set of robotic manipulation tasks for benchmarking multitask-learning and meta-learning. In this experiment, we consider the MT10 benchmark (Appendix C.1.4) from Meta-World, where we have 10 diverse tasks, evaluated with the mean success rate over 10 tasks.

We run our experiments on MT10 with 10 seeds for 100k gradient updates for each task. We evaluate our trained agent 100 times, which gives a more accurate performance, on all MT10 tasks. We report

Table 4: Performance comparison of Meta-World online. We compare the success-rate of NPF on MT10 online. We also show the *reduced parameter complexity* of NPF comparing to other baseline methods.

| Experiments | SAC+**NPF** | PCGrad Yu et al. (2020) | SM Yang et al. (2020) | CARE Sodhani et al. (2021) | SAC+ME Sodhani et al. (2021) |
|---|---|---|---|---|---|
| MT10 tasks | **77.4 ± 13.12** | 72.0 ± 2.2 | 73 ± 4.3 | 84 ± 5.1 | 74 ± 4.3 |
| Parameter Counts | 17k | 340k | 135k | 486k | 344k |
| Parameter Reduction | - | 20x | 8x | 28x | 20x |

the mean performance with 95% confidence interval in Table 3 (see individual task performance in Table 7 in Appendix). We collect 1k expert trajectory (150k samples) per task using SAC (Haarnoja et al., 2018) expert trained on a single task. Using only 5% of the network weights, neural-pathways is able to exceed other baselines. Whereas other baselines suffer from high variance in performance, with BCQ, we get a perfect score on all MT10 tasks.

## 5.2 ONLINE MULTITASK PERFORMANCE EVALUATION

A recent work (Sodhani et al., 2021) benchmarks state-of-the-art online multitask RL algorithms in the Meta-World environments, shown in Table 4. The benchmark is reported after 2M gradient updates (for each environment) over seeds 0-9. We follow their evaluation criterion to report success rate of our algorithm. Sodhani et al. (2021) focuses on the importance of contextual information and proposes *Contextual Attention-based Representation* (CARE), using a pretrained language model (Liu et al., 2019) to encode meta-data and retrieve contextual information about the task. The contextual information added with the environment observation is used to train *SAC with Mixture of encoders* (SAC+ME). As seen in Table 4, the success of CARE largely depends on metadata which is not always available in all multitask settings. *Soft-modularization* (SM) proposes effective sharing and reusing network com-

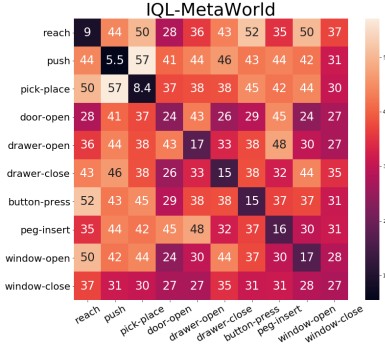

Figure 6: The % of trainable weights the IQL policy shares across MT10 tasks with NPF. The diagonal is the % of weights unique to each task.

ponents across tasks. As reported in Sodhani et al. (2021), the performance of SM (Yang et al., 2020) varies largely when evaluated over 10 different sub-tasks. We also include *Projecting Conflicting Gradient* (PCGrad, Yu et al. (2020)), which focuses on tackling gradient interference. If gradients for different tasks point away from one another, PCGrad alters gradient direction to mitigate these interference.

Compared to these baseline methods, we propose a simpler alternative that achieves competitive performance while not requiring additional task information, complicated network architecture such as routing through modular networks, or a complex gradient update procedure. We train *each task* with a fraction of the parameters (see Table 4) and which potentially leads to higher inference (Molchanov et al., 2017; Luo et al., 2017) and reduces the energy consumption (Yang et al., 2016), which makes it more suitable than any other multitask methods to apply in real-world applications.

## 5.3 OVERLAPPING NEURAL PATHWAYS

In this work, we consider choosing the neural pathways for each task independently from one another. We find a large % of the pathways overlap, reducing the number of trainable parameters. The pathway configuration is dictated by three factors: (1) different learning objectives (i.e. offline, online), (2) scoring function (i.e. equation 1, 2), and (3) training samples used to optimize the scoring function. In Figure 6 we see the % of active weights (5% of the actual network) in the policy that are shared with other tasks for IQL multitask-offline training. The diagonal of these symmetric matrices represent the number of unique weights that are optimized only for one task, and the columns represent the % of weights each task shares with others. As we see in Figure 6, for all tasks the % of weights that are optimized for just one task is very low. For *push*, only 5.5% of the *active weights* are uniquely trained for the task. We also show similar matrices for other experiments in Appendix (Figure 15), where we found the percentage of overlapping weights does not correlate with the task similarity. This disproves a conventional class of thinking that the success of multitask in neural networks depends on relevant data sharing (Yu et al., 2021; Fifty et al., 2021) or similar task training (Sodhani et al., 2021;

Yang et al., 2020). Rather, neural networks are capable of learning multiple tasks simultaneously as long as the neurons are wired properly.

**Overlapping Pathways Lead to Fewer Parameters:** For each task we only use $5\%$ of the total weights of the actual network while maintaining expert-like performance for all tasks. For $N$ tasks it has a upper bound of $\frac{N}{20}x$ parameters when all the pathways are unique, but we find the pathways to overlap significantly (Figure 6 and Figure 15, Appendix), further reducing the number of active weights. This reduces the energy consumption (Yang et al., 2016) of the trained network and makes inference more efficient (Molchanov et al., 2017; Luo et al., 2017). To understand how NPF compacts multiple experts into single network, we look at the number of policy parameters IQL requires compared to SAC with and without pruning on the individual MT10 tasks. In our experiments, the total number of network parameters for SAC and IQL are identical with $1,092,608$ parameters in the policy network. With MT10 the number increases $10\times$. Due to the inherent shared structure of neural pathways, our method requires only $1.82\%$ (averaged over 10 seeds) of the original network. To get equivalent utility we have to train ten SAC agents with $98.18\%$ pruned networks, which is not possible using any existing pruning techniques without losing performance (Frankle & Carbin, 2019; Lee et al., 2018; Wang et al., 2020a; Tanaka et al., 2020). As seen in Wang et al. (2020a) at $98\%$ reduction of the network weights pruning techniques suffers from significant performance drop due to layer collapse. NPF reduces the energy consumption (Yang et al., 2016) of the trained network, making inference more efficient (Molchanov et al., 2017; Luo et al., 2017). See the comparison in Appendix (Table 8).

## 6 CONCLUSION

We propose a novel multitask learning approach in which we train task-specific neural pathways within a single deep neural network. We extend the idea of finding sub-networks from the pruning literature (Frankle & Carbin, 2019; Wang et al., 2020a; Feinberg, 1982) and show our proposed method can achieve expert-like performance on many high dimensional continuous control tasks in both online and offline settings. Unlike conventional approaches, where multitask methods mostly rely on task similarity, modular network training, or underlying context learning, we propose a new way to tackle multitask problems and show function approximation with a single deep neural network is sufficient. We empirically show that the *single-shot lottery network search* fails under data distribution shift We overcome this by proposing *data adaptive pathway discovery*, which continues to reconfigure neural pathways as the RL agent gets more training updates and continues to collect better training samples. Leveraging offline and online RL algorithms, we train task-specific neural pathways in parallel, consisting of only $5\%$ of the total neural network weights, leading to significant reductions in training time while still achieving competitive performance on Meta-World. Due to significant neural-pathway overlap, we get a further reduction in network parameters. Compared to single-task experts, we get $\geq 98\%$ pruned parameters in the Meta-World experiments, a significant decrease in parameters which is not possible for single-task training with any existing pruning techniques without a performance drop. Thus, NPF can provide a high inference rate while reducing energy consumption (Yang et al., 2016), making it more very suitable for real-world applications.

This work opens up a new avenue of inquiry into the possibility of a single neural network being trained for multiple purposes (i.e. learning different features, multiple value functions) through targeted parameter training while providing better efficiency of utilizing the parameter space of neural networks. Our method can be integrated into other selective data-sharing and gradient update methods but this requires further research. Expanding the scope of the experiments to include more tasks would also be informative.

## 7 REPRODUCIBILITY:

With this work we provide an open-source implementation of our framework at https://github.com/anomICLR2023/2904. Users can run NPF and reproduce the experiments utilizing the code in the anonymous github repo.

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

# A   ADDITIONAL THEORETICAL BACKGROUND

## A.1   OFFLINE AND ONLINE THEORY

Fujimoto et al. (2018) highlight the fact that, since value estimation is trained with fixed dataset, it provides an erroneous estimation when the policy takes an action which is out of distribution from the dataset on which the value function is trained. To overcome extrapolation error, BCQ proposes a batch-constrained learning, where agents are trained to maximize reward while minimizing the mismatch between the state-action visitation of the policy and the state-action pairs contained in the batch. For a given state, BCQ uses a generative model $G_w$, e.g. a Variational Auto-encoder Kingma & Welling (2014a), to generate $n$ actions with high similarity to the batch dataset, and then it selects the action for which it gets the highest value: $\pi(s) = \arg\max_{A_i} Q_\phi(s, a_i)$, where $A_i \sim \{G_w(s)\}_{i=1}^n$. $G_w$ is trained to minimize the KL divergence with the actions sampled from batch dataset Kingma & Welling (2014b). The action-value function or $Q$-function, $Q_\phi$ learned through minimizing TD-error is $J(Q_\phi) = \mathbb{E}_{\{s,a,s'\} \sim D, a' \sim \pi_\theta}[(r(s,a) + \gamma \hat{Q}_\phi(s', a')) - Q_\phi(s, a)^2]$. In our experiments, we also consider a variation of BCQ (denoted as BCQ-v2) in which we sample actions only using a VAE ($\pi(s) = G_w(s)$).

Since Offline RL faces distribution shift in its value estimation due to the different distribution of he policy and expert sample, *Implicit Q-learning* (IQL) Kostrikov et al. (2021) proposes to avoid estimating the value for policy distribution. Instead, it trains the action-value function $Q_\phi$ using a SARSA-style update, thus enabling multi-step dynamic programming updates. IQL uses expectile regression to predict an upper expectile of TD target that approximates the maximum of $r(s,a) + \gamma[\hat{Q}_\phi(s', a')]$. IQL uses a separate value function by fitting upper expectile $V_\psi$ using objective function: $J(V_\psi) = \mathbb{E}_{s,a \sim D}[\mathcal{L}_2^\tau(\hat{Q}_\phi(s', a') - V_\psi(s))]$, where $\mathcal{L}_2^\tau$ is asymmetric least squares. This value is used to update $Q$ function using: $J(Q_\phi) = \mathbb{E}_{s,a,s',a' \sim D}[(r(s,a) + \gamma \hat{V}_\psi(s') - Q_\phi(s,a))^2]$. The corresponding policy is extracted using advantage weighted behavior cloning, which also avoids querying out-of-sample actions:$J(\pi_\theta) = \mathbb{E}_{s,a,s',a' \sim D}[(Q_\phi(s,a) - V_\psi(s)) \log \pi_\theta(a|s)]$.

We consider no prior knowledge about the task in the online RL setting. An RL agent is expected to interact with a simulated environment and leverage the collected experience to learn the task through maximizing a hand-designed reward function. Neural pathways can be integrated into any online RL algorithm and trained for divergent multitask objectives. To demonstrate the effectiveness of our method in the online setting, we use *Soft-Actor-Critic* (SAC) Haarnoja et al. (2018) algorithm in our experiment. SAC optimizes entropy-regularized policy objectives to drive an agent to a more exploratory behavior while optimizing its policy. Entropy is used to encourage policy to do more exploratory behavior and ensure that it does not collapse into repeatedly selecting a particular action. The entropy regularized objective function is as follows:$J(\theta) = \sum_{t=0}^T \mathbb{E}_{s_t \sim d_\pi, a_t \sim \pi_\theta}[r(s_t, a_t) + \alpha \mathbb{H}(\pi_\theta(.|s))]$, here $\alpha$ is *temperature parameter*, which controls the *entropy*, $\mathbb{H}$ parameter, hence the stochasticity of the policy. It determines the relative importance of the entropy term against the reward. The conventional objective for policy gradient is recovered when $\alpha \to 0$. SAC learns a $Q$-function using an entropy regularized objective called soft $Q$-function. The soft $Q$-function parameters are trained to minimize the following objective:$J(\phi) = \mathbb{E}_{(s,a) \sim D}[(Q_\phi(s_t, a_t) - r(s_t, a_t) + \gamma \mathbb{E}_{s \sim \rho^\pi, a \sim \pi_\theta}[Q_{\phi'}(s_{t+1}, a_{t+1}) - \alpha \log \pi_\theta(a_{t+1}|s_{t+1})])^2]$.

# B   ADDITIONAL IMPLEMENTATION DETAILS

## B.1   LIBRARIES

We run our algorithm in PyTorch-1.9.0 Paszke et al. (2019) and use following libraries: Soft-Actor-Critic (SAC) Yarats & Kostrikov (2020), Implicit Q-learning (IQL) Thomas (2021), Single-shot pruning (SNIP) Alizadeh (2019) and official BCQ Fujimoto et al. (2018) implementation.

## B.2   OFFLINE DATA COLLECTION

We use the Halfcheetah control environment available from Thomas (2020) and train Soft-Actor-Critic (SAC) Haarnoja et al. (2018) online for 1 million time steps for each task and collect 1000 expert trajectory.

For Quadrupod tasks we use the quadrupod environment and trained agents from Smith et al. (2021) and collected 1000 trajectories for each task. During data collection we find it important to use

stochastic policy to add data diversity, otherwise every trajectory follows the exact consecutive states,actions and rewards sequence due to the deterministic nature of the environment transition function.

We train Soft-Actor-Critic (SAC) Haarnoja et al. (2018) on Meta-World Yu et al. (2019) environments for 3 million steps, where sample a batch of 1024 sample and the hidden dimension is 1024. Similar to Sodhani et al. (2021) we truncate the episode for 150 steps. We collected 1k trajectories for each environment, where we take sample action of the normal distribution rather than the mean to have diversity in the training sample. We keep the environment seed fixed at 0, since Meta-World Yu et al. (2019) goal position is different for different seeds. Since we are not learning a goal-conditioned algorithm, we need to keep this seed fixed other wise for each seed we have to train and collect expert data for offline training.

### B.3 HYPER-PARAMETER

In table 5 we present the network hyper-parameters of different algorithms that are used in this work. We make exception in MetaWorld online experiments, where we use mini-batch size 128 and optimizer AdamW.

Table 5: Hyperparameter of the network architecture used to train and evaluate offline and online RL experiments.

|  | Hyper-parameter | BCQ | IQL | SAC |
|---|---|---|---|---|
| hyper-parameter | Optimizer | Adam | Adam | Adam |
|  | Critic learning rate | 1e-3 | 1e-3 | 1e-3 |
|  | Actor learning rate | 1e-3 | 1e-3 | 1e-3 |
|  | Mini-batch size | 256 | 256 | 256 |
|  | Discount factor | 0.99 | 0.99 | 0.99 |
|  | Target update rate | 5e-3 | 5e-3 | 5e-3 |
|  | Policy update frequency | 2 | 2 | 2 |
| Architecture | Critic hidden dim | [400, 300] | [1024, 1024] | [400, 400, 400] |
|  | Critic activation function | ReLU | ReLU | ReLU |
|  | Actor hidden dim | [400,300] | [1024, 1024] | [400, 400, 400] |
|  | Actor activation function | ReLU | ReLU | ReLU |
|  | VAE hidden dim | [750, 750] | – | – |
|  | Value hidden dim | – | [1024, 1024] | – |

### B.4 PERFORMANCE EVALUATION

Each task in a multitask experiment is trained with an equal number of gradient updates. We evaluate performance every 5000 gradient updates, where each evaluation reports the average episodic return over 10 episodes. Our results are reported over multiple seeds of the simulator and the network initialization.

We evaluate our agent on average episodic return of 10 episodes. For offline experiment, we normalize the episodic return using standard proposed metric from Fu et al. (2020): Normalized score $= \left( \frac{\text{score - random score}}{\text{expert score - random score}} * 100 \right)$. We plot the mean normalized score of multiple seed with $100\%$ confidence interval for all our experiments. For MetaWorld (in both offline and online) we evaluate the percentage of success.

For Gym environment we conduct experiments for seed 0-4 (if not stated otherwise). We run all MetaWorld experiments for seed 10 seeds (0-10) and report the percentage of success after evaluating for 100 episodes.

## C ADDITIONAL RESULTS

### C.1 OFFLINE MULTITASK EXPERIMENTS

We provide the individual task performance compared to *single task expert* for `Halfcheetah` and `Quadrupod` multitask experiments in Table 6.

Table 6: Individual task normalized score (performance compared to *single task expert*) `HalfCheetah` and `Quadrupod` tasks.

| Experiment | Tasks | NPF | |
| --- | --- | --- | --- |
| | | BCQ | IQL |
| `HalfCheetah` multitask | Forward | 99.68 ± 0.04 | 99.98 ± 0.04 |
| | Backward | 96.51 ± 0.19 | 98.68 ± 0.83 |
| | Jump | 71.38 ± 9.46 | 74.99 ± 20.1 |
| | Forward-Jump | 85.25 ± 0.54 | 87.35 ± 27.0 |
| | Backward-Jump | 83.3 ± 1.46 | 87.34 ± 14.23 |
| `HalfCheetah` Constrained speed | Velocity 0.5 | 100.04 ± 0.03 | 99.1 ± 2.18 |
| | Velocity 1.0 | 100.07 ± 0.03 | 100.05 ± 0.04 |
| | Velocity 1.5 | 100.03 ± 0.05 | 99.99 ± 0.06 |
| | Velocity 2.0 | 100.07 ± 0.01 | 100.07 ± 0.04 |
| | Velocity 2.5 | 100.02 ± 0.04 | 100.04 ± 0.03 |
| | Velocity 3.0 | 99.96 ± .07 | 100.01 ± 0.03 |
| `Quadrupod` multitask | Forward | 116.19 ± 0.14 | 116.92 ± 0.51 |
| | Backward | 110.98 ± 1.21 | 112.93 ± 1.05 |
| | Hopturn | 122.43 ± 0.47 | 123.33 ± 0.29 |
| | Sidestep | 111.8 ± 0.96 | 113.69 ± 0.43 |

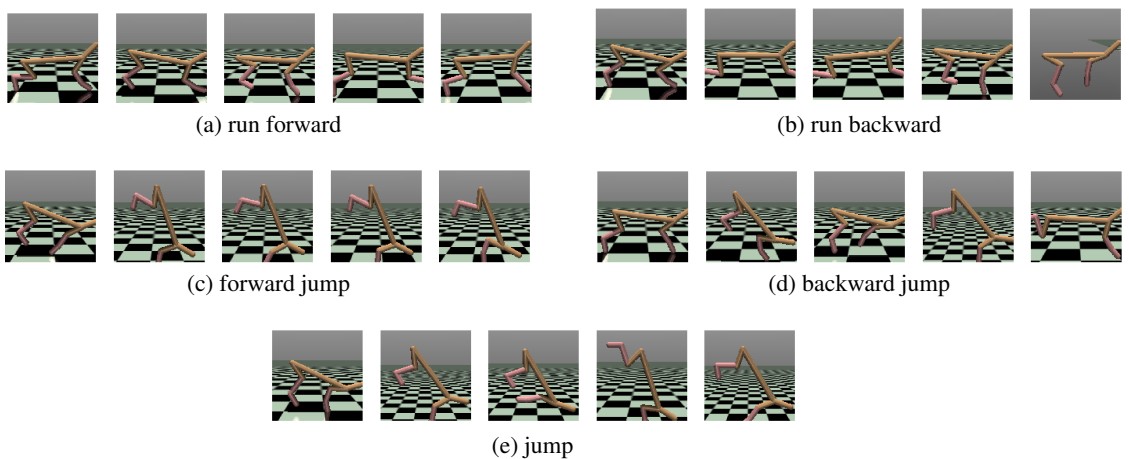

(a) run forward        (b) run backward

(c) forward jump        (d) backward jump

(e) jump

Figure 7: Snapshot of trained policy for `Halfcheetah` multitasks.

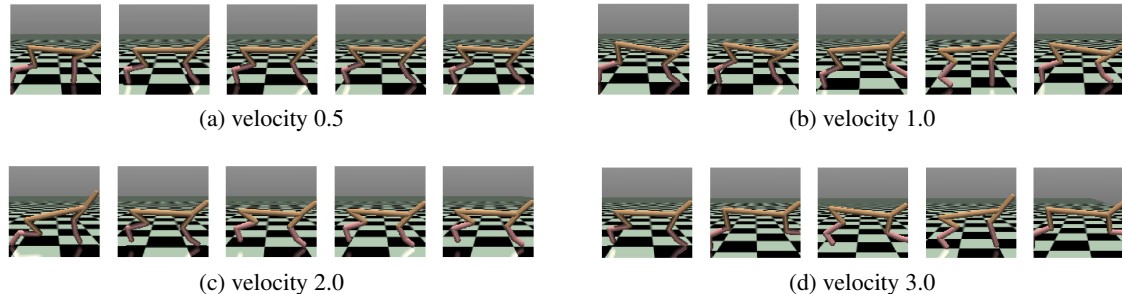

(a) velocity 0.5        (b) velocity 1.0

(c) velocity 2.0        (d) velocity 3.0

Figure 8: Snapshot of trained policy for `Halfcheetah` under four different constrained velocity.

### C.1.1 PERFORMANCE CURVE OF HALFCHEETAH MULTITASK

Figure 10 presents the performance of five different `Halfcheetah` tasks: (1) run forward, (2) jump while run forward, (3) run backward, (4) jump while run backward and just(5) jump. Snapshots of the different tasks are demonstraded in Figure 7.

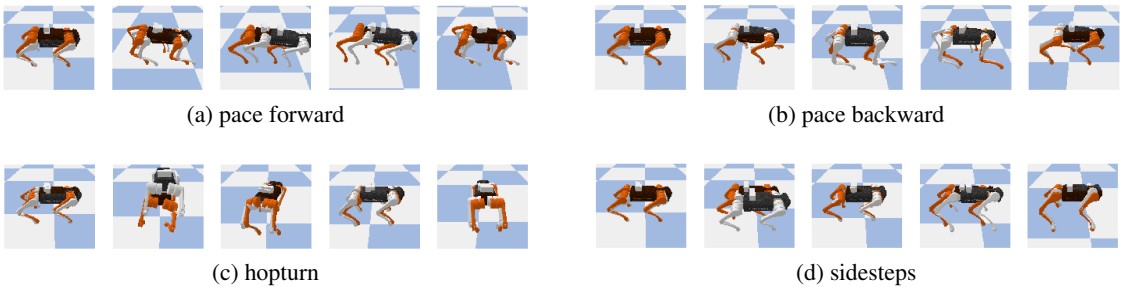

(a) pace forward           (b) pace backward

(c) hopturn           (d) sidesteps

Figure 9: Snapshot of trained policy for `Quadrupod` multitasks.

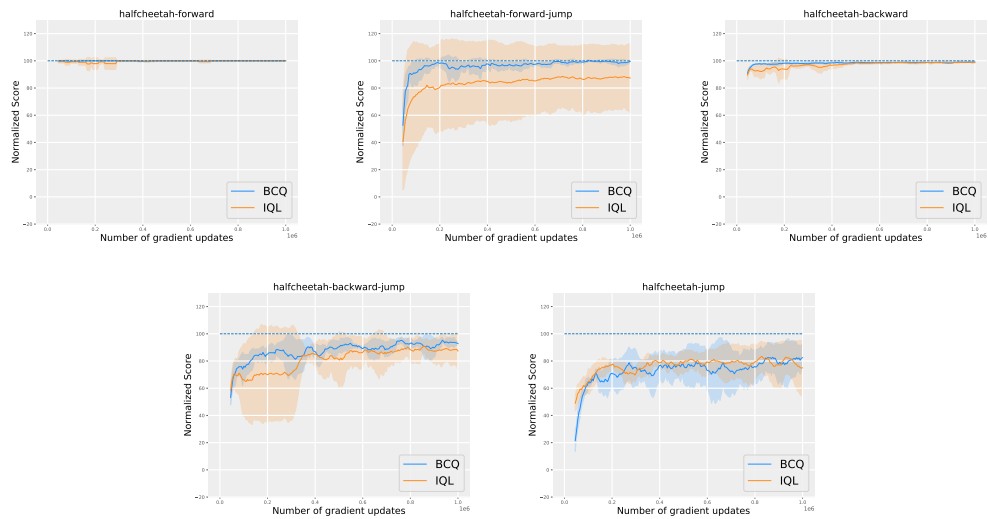

Figure 10: Performance (Normalized Score) plot of BCQ and IQL on `HalfCheetah` multitask - (Forward, Forward-Jump, Backward, Backward-Jump, Jump) trained with NPF.

### C.1.2 PERFORMANCE CURVE OF HALFCHEETAH CONSTRAINT GOAL VELOCITY

In Figure 11 we present the performance of six (6) constrained goal velocity for `Halfcheetah` running forward, where we vary the constrained from $0.5$ to max speed $3.0$ with a constant increase of the speed. The tasks are trained for 500k gradient updates and evaluated every 5000 gradient updates. Snapshots of the different tasks are demonstraded in Figure 8.

### C.1.3 PERFORMANCE CURVE MULTITASK QUADRUPED ROBOT

We use the `Quadruped` robot to perform four (4) tasks: hopturn, pace forward, pace backward, sidestep. We use Smith et al. (2021) as our expert to collect 500k (1000 trajectory) expert data for each task. The environment has a deterministic transition function and thus we take sample action from normal distribution instead of taking distribution mean to get a diverse set of training dataset. Performance curve with NPF for individual task in shown in Figure 12. Snapshots of the different tasks are demonstraded in Figure 9.

### C.1.4 PERFORMANCE ON METAWORLD

In Table 7 we report mean performance of individual MT10 environments over 10 seeds and each evaluated for 100 episodes. We also compare individual performance with *Conservative data sharing* (CDS) Yu et al. (2021), an offline-multitask learning algorithm that focuses on training based on task similarity, also reports it's Meta-World experiments on 4 environments and reported performance over 6 seeds with $95\%$ confidence interval.

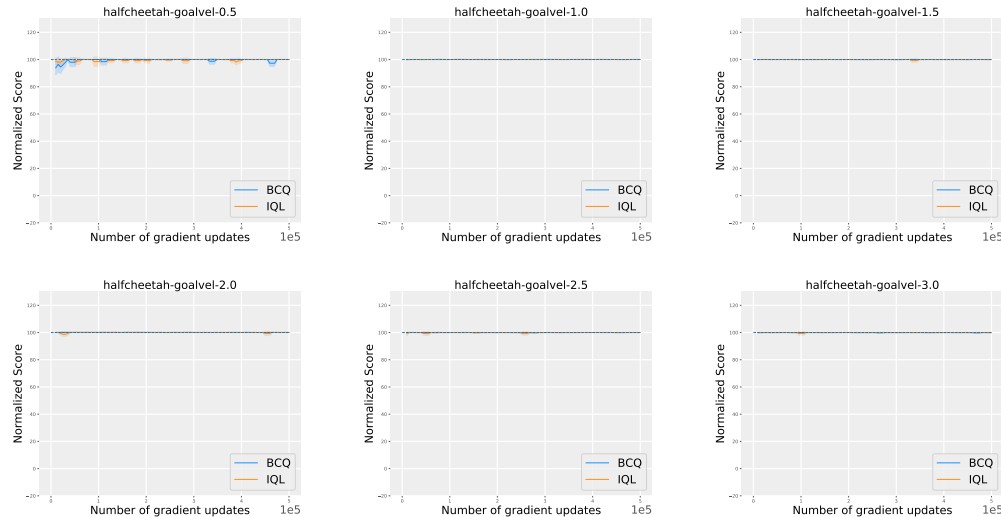

Figure 11: Performance (Normalized Score) plot of BCQ and IQL on `HalfCheetah` with constrained velocity trained with NPF.

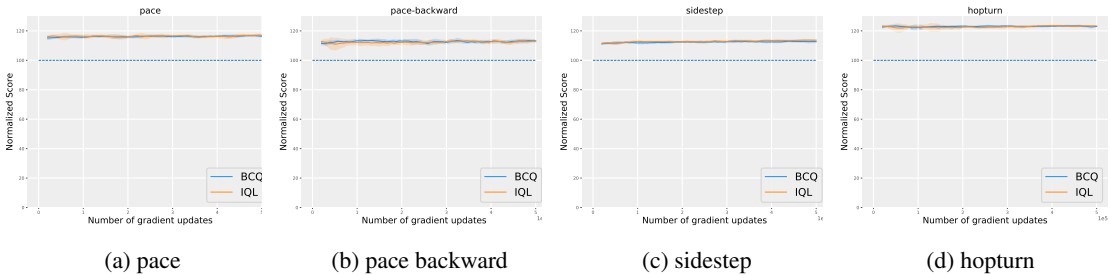

(a) pace      (b) pace backward      (c) sidestep      (d) hopturn

Figure 12: Performance (Normalized Score) plot of BCQ and IQL on `Quadrupod` multitasks with NPF.

Table 7: Individual task performance (success-rate) comparison of Meta-World.

| MT10 | NPF | | CDS Yu et al. (2021) |
|---|---|---|---|
| | BCQ | IQL | |
| reach | $100.0 \pm 0.0$ | $93.0 \pm 22.14$ | - |
| push | $100.0 \pm 0.0$ | $100.0 \pm 0.0$ | - |
| pick-place | $100.0 \pm 0.0$ | $95.0 \pm 15.81$ | - |
| door-open | $100.0 \pm 0.0$ | $100.0 \pm 0.0$ | $58.4 \pm 9.3$ |
| drawer-open | $100.0 \pm 0.0$ | $85.0 \pm 33.75$ | $57.9 \pm 16.2$ |
| drawer-close | $100.0 \pm 0.0$ | $100.0 \pm 0.0$ | $98.8 \pm 0.7$ |
| button-press-topdown | $100.0 \pm 0.0$ | $100.0 \pm 0.0$ | - |
| peg-insert-side | $100.0 \pm 0.0$ | $100.0 \pm 0.0$ | - |
| window-open | $100.0 \pm 0.0$ | $100.0 \pm 0.0$ | - |
| window-close | $100.0 \pm 0.0$ | $100.0 \pm 0.0$ | - |
| door-close | - | - | $57.2 \pm 16.2$ |
| Overall | $\mathbf{100 \pm 0.0}$ | $\mathbf{97.3 \pm 7.17}$ | $72.0 \pm 26.2$ |

## C.2 ONLINE MULTITASK EXPERIMENTS

We provide the pseduo-code for *Data Adaptive Pathway Discovery* (DAPD) proposed for online multitask setting in Algorithm 2.

---

**Algorithm 2** Multitask Online Training

---

▷ Episodic Return $n^{th}$ task, $R_T^n = \sum_{t=1}^{T} r_t^n$; Stopping threshold $T_H$: an hyper-parameter; Temporary replay buffers $n^{th}$ task, $TD_n$: only contains most recent experience; Replay buffers $n^{th}$ task, $D_n$; Mask for $n^{th}$ tasks, $m_n$.

#Training loop:

**for** Training steps **do**

   # Collect Data:

   **for** Each Task **do**

      ▷ $a_t \sim \pi_\theta(a_t|s_t)$, $s_{t+1} \sim T_n(s_{t+1}|s_t, a_t)$

      ▷ $D_n \leftarrow D_n \cup \{s_t, a_t, r(s_t, a_t), s_{t+1}\}$

      ▷ $TD_n \leftarrow TD_n \cup \{s_t, a_t, r(s_t, a_t), s_{t+1}\}$

   **end for**

   # Find the Task Specific Weight Masks:

   **for** Each Task **do**

      ▷ **If not** $R_T^n \geq T_H$ : $m_n$ = FindNeuralPathways $\left(TD_n\right)$; using Equation 2.

   **end for**

   # Update Policy using Online Algorithm:

   ▷ Update $\pi_\theta$ ( $\{m_1, m_2 .. m_n\}$, $\{D_1, D_2 .. D_n\}$) # Follow Same Approach as Algorithm 1 for weight and gradient share among tasks

**end for**

---

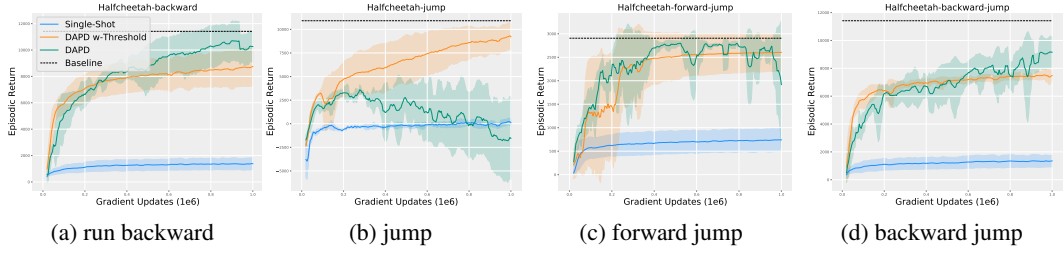

| (a) run backward | (b) jump | (c) forward jump | (d) backward jump |

Figure 13: Compare Performance of Single-Shot and *Data adaptive Pathway Discovery* (DAPD with and without Threshold Stopping Criterion) on single Mujoco tasks.

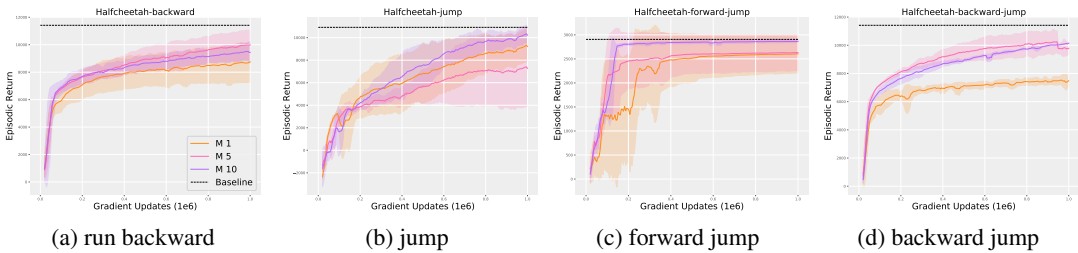

| (a) run backward | (b) jump | (c) forward jump | (d) backward jump |

Figure 14: Compare Performance of *Data adaptive Pathway Discovery* (DAPD) with different moving average on single Mujoco tasks.

### C.3 OVERLAPPING NEURAL PATHWAYS

**Important Distinction with Conventional Pruning Methods:** Neural Pathway has a crucial technical difference with pruning methods in the implementation. As demonstrated in Figure 2, we mask the neural network weights, allowing forward and backward propagation only through a subset of weights, whereas pruning methods Feinberg (1982); Wang et al. (2020a); Frankle & Carbin (2019) set many weights to zero. This technique allows the unimportant weights to survive and be used for other tasks. In general, pruning methods are used to reduce memory footprint, increase throughput

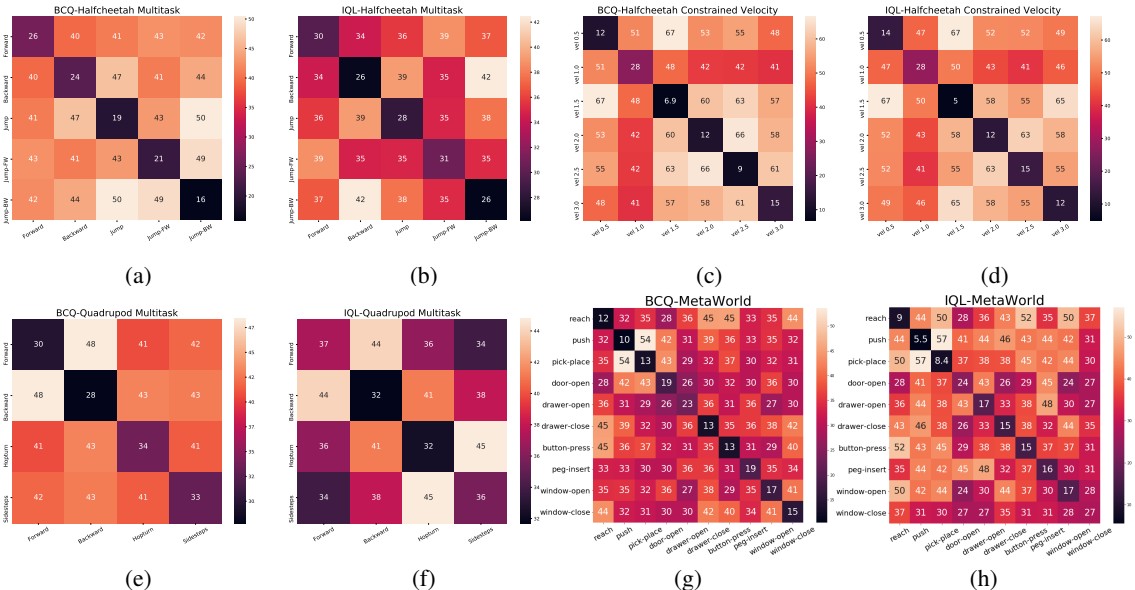

Figure 15: Shows the percentage of trainable weights are being shared among different tasks. In $a$ and $b$, we compare shared weights among five `Halfcheetah` tasks: (1) forward (F), (2) backward(B), (3) jump(B), (4) Forward while Jumping (FJ) (5) Backward Jumping (BJ). In $c$ and $d$, we compare shared weights among six `Halfcheetah` tasks where we increase forward velocity 0.5-3.0 . In $e$ and $f$, we compare four `Quadrupod` tasks. In $g$ and $h$ we show the shared weights among MetaWorld tasks.

or even enhance generalization. On the other hand, our goal is not to prune but to re-purpose the capacity of the network to learn many tasks (to a larger extent than in other works).

In Figure 15 we demonstrate the percentage of active weights (5% of actual network) in policy network that are shared with other tasks for different experiments and for different offline RL algorithms. The diagonal of these symmetric matrices represent the number of unique weights that are optimized only for one task. And the columns represents the percentage of weights the task shares with others.

It is important to note, in Figure 15, for all our experiments the percentage of weights that are optimized for just one task is very low. For examples, in Figure 15-$a$ for forward task, only 26% of the active weights are uniquely trained for the task. We only activate/allow 5% of the original network weights for one task after pruning. Which means only 0.13% of the original network is trained uniquely to make Halfcheetah run forward like an expert. Also we do not find the percentage of overlapping to correlate with the tasks similarity.

In Table 8, we compare the number of policy network parameters IQL requires compared to SAC with and without pruning when trained for MT10 environments as individual tasks. In our experiments, the network hidden layers and depth (total number of network parameters) for SAC and IQL are identical. Our SAC expert has $1,092,608$ weight parameters in the policy network. For MT10 (10 tasks), the number increases $10x$. On the other hand, due to the inherent shared structure of neural pathways, our method requires only 1.82% (in average over 10 seeds) of the original network. To get equivalent utility we have to train ten SAC agents with 98.18% pruned network, which is not possible using any existing pruning techniques without losing performance Frankle & Carbin (2019); Lee et al. (2018); Wang et al. (2020a); Tanaka et al. (2020). As seen in Wang et al. (2020a) at 98% reduction of the network weights pruning techniques suffers from significant performance drop due to layer collapse. But in our case, the more tasks we have the more space efficient our method becomes. Even when we compare to 95% pruned SAC trained on 10 different individual tasks ((B) in table 8), we have 36.4% reduction in actor parameters.

Table 8: Comparison of the number of parameters required in Meta-World for different methods.

| | MT-10 trained as separate tasks | | MT-10 multitasks trained using **NPF** |
|---|---|---|---|
| | (A) w/o pruning | (B) 95% pruned | (C) 95% pruned |
| Parameter counts (Policy/Actor Network) | 11 Million (10,926,080) | 546,304 | $198,956 \pm 7,636$ ( $1.82 \pm 0.07\%$ of A) |

