# OpenReview forum: "Multitask Reinforcement Learning by Optimizing Neural Pathways"
_ICLR.cc/2023/Conference — Submitted to ICLR 2023_

### Official Review · Reviewer_rc1X · 2022-10-24

**Confidence:** 4
**Correctness:** 3
**Technical Novelty And Significance:** 2
**Empirical Novelty And Significance:** 2
**Recommendation:** 3

**Clarity, Quality, Novelty And Reproducibility:**

- This paper could be improved by clearly showing more details of the proposed method as I mentioned in the main Weaknesses. Besides, there are some questions that could be clarified.
- The idea of finding neural pathways with pruning is not new. Since the current paper is not organized and written in a good form, especially on the methodology part, it's hard for me to evaluate its originality accurately.

**Strength And Weaknesses:**

Strengths:
- Multitasking is an important ability to achieve artificial general intelligence (AGI). This is an interesting work targets to multitask RL where the ideas build up from the neuro-science perspective on neural pathways in human brains and the machine learning perspective on finding sub-networks with pruning.

Weaknesses:
- The proposed methods are not presented in a principled way, although the main contribution of this work is primarily on the empirical side. For example, what's the formal definition of the problem? What are the loss functions or objectives for the offline or online settings? What's the mask used in the methods? What are the original networks and sub-networks? How do neural pathways/sub-networks interact with different tasks? How to guarantee or justify that the obtained sub-network is the right solution? How are the learned skills reused/transferred among tasks? What does the mask "$m$ is fixed" mean?
- In Figure 4 (a), why the gradient norm becomes larger? Does that mean there exists a convergence issue? How to define the success rate in (b)?
- This paper claims that the proposed method "is especially suitable for real-world applications in which large datasets need to be processed in real-time or which are deployed on low-resource devices". However, I didn't see any comparisons of time cost or computation cost shown in the experiments.


Minors:
- "In the online RL setting,." -> "In the online RL setting,".
- "nonstationaritym" on pp.5 should be "nonstationarity".

**Summary Of The Paper:**

This paper focuses on multitasking ability in reinforcement learning (RL). To do so, a multi-task learning framework, Neural Pathway Framework (NPF), is proposed by simultaneously training multiple specialized pathways where each pathway corresponds to a task. Finally, experiments were conducted on several continuous control tasks under both online and offline settings, including using Mujoco-based and MetaWorld environments, with results demonstrating its effectiveness in terms of performance and parameter size.


**Summary Of The Review:**

According to my comments in both the main Weaknesses and the section of "Clarity, Quality, Novelty", I feel this is an empirical paper where reasons to reject outweigh reasons to accept.

---

> ### Author Response · Authors · 2022-11-19
> **Authors' Response to Reviewer rc1X (Part 2/2)**
>
>
> > ***What does the mask "m is fixed" mean?***
>
> By "the mask, m is fixed" (Section 4.1) we mean, we have decided not to change/update the mask. We explain the purpose of using mask in Section 4.1. A mask, m allows us to forward and backward propagate specific sets on neural network parameters (pathway/subnetwork). Once we discover a task-specific pathway (offline -algorithm 1, online algorithm 2), we keep the mask fixed allowing us to train the task-specific pathways.
>
> * The reviewer also pointed out in Figure 4(a) “the gradient norm become larger” when plotted across gradient updated and asks “there exists a convergence issue?”
> A larger "gradient norm" of parameters indicates its larger change from the randomized initial value over time. This is a good sign since parameters are getting updated from random initialization.
> The figure (4a blue ) does not necessarily mean, it has a convergence issue. In the ideal case, we do expect the gradient norm eventually to decrease as the parameter converges but as we see in (Figure 4b blue), the harder tasks in MT10 do not converge to a perfect score (success-rate =1.0 on the scale of 1 ) after 2 million gradient updates
>
> > ***“How to define the success rate in (b)?”***
>
> Metaworld tasks are evaluated based agent's success to reach an expected goal state, which is defined in [2]. We added the explanation of the evaluation criteria on the updated version of the paper.
>
> * **NPF compared to other baselines in real-world application:**
>
> The reviewer asks a “comparisons of time cost or computation cost” to support our claim that the NPF method is more favourable to “real-world applications”.
>
> We support our claim by showing that the NPF requires the lowest floating point operation (FLOP) counts when compared to offline (Table 3) and online (Table 4) baselines. A lower FLOP count is linearly proportional to faster inference. For any given task we use [1] to compute FLOP for each method. To completely exploit the sparsity adaptation of prevalent deep learning framework is needed at a much lower level. However, with latest nvidia GPUs, they can already take advantage of sparsity in parameters [3].
>
> > ***“The idea of finding neural pathways with pruning is not new”***
>
> We want to point out that pruned/sparse neural network applied in multitask setting in novel to our paper and we show its significance in both offline and online settings.
> We agree with the reviewer that the idea to find sub-network (single) with pruning is indeed not new and we did not claim it as such. However, we believe the idea to leverage the subnetworks (multiple) found from scoring mechanisms that were traditionally used for pruning and use multiple somewhat overlapping sub-networks to solve multi-task problem is novel. In this paper, we define neural pathways as different subsets of a single neural network and their purpose to learn different task is novel to our paper.
>
> [1] https://github.com/facebookresearch/fvcore
> [2] Meta-World: A Benchmark and Evaluation for Multi-Task and Meta Reinforcement Learning
> [3] Nvidia. How sparsity adds umph to ai inference. 2020.

---

> ### Author Response · Authors · 2022-11-19
> **Authors' Response to Reviewer rc1X (Part 1/2)**
>
> We thank the reviewer for their time in reviewing the paper and providing feedback. We are enthused that the reviewer finds the connection between “finding sub-networks with pruning” and “neuroscience perspective on neural pathways in human brains” interesting.
> The reviewer mostly concerns with the clarity of the proposed method and raises important questions. But we argue that most of the reviewer's concerns are already discussed in detail in the paper. Here and in the updated version of the paper, we add further clarification.
> Clarification on the proposed method:
>
>
> >***“What are the loss functions or objectives for the offline or online settings?”***
> >
> The proposed method to find pathways using pruning is agnostic to RL algorithm, and we show this by combining it with different online and offline RL algorithms. The loss/ objective function for offline or online setting are subject to change for different algorithms. In this paper, we experimented with IQL and BCQ in offline and SAC in online settings, Which are provided in Appendix A1. NPF can be integrated without requiring any changes in the loss function or objective function
>
> > ***“What's the mask used in the methods?”***
>
> We introduce the mask in Section 4.1 (paragraph 2):
>  “Formally, we consider a feed-forward network f (x, θ) with initial parameters θ0 ∼ Dθ . A Neural Pathway activates a sub-network f (x, θ ⊙ m) by using a mask m ∈ {0, 1}|θ|)”
>
> We discussed how it allows training for multiple tasks ( illustrated in Figure 2) in Section 4.2:
> “Training a Neural Pathway: First, we identify the top 5% most important weights for each task independently using Equation (1), which will be remembered using binary masks, one for each task, in which 1 marks trainable weights. We never set any weights to zero. But, due to masking, only the subset of the weights contained in the neural pathway participate in task-specific inference and updates. Thus, for any given task, the mask mn activates a pathway of neurons for forward propagation and backward propagation) “
> And discussed further in detail how the mask m is configured using dataset D and mask allow task specific forward and backward propagation in “Simultaneous Multitask Training” in Section 4.2
>
> > ***“What are the original networks and sub-networks?”***
>
> We address the concept of sub-network Section 2:
> “Recent advances in finding sparse networks have proven that there exist sub-networks which contain a small fraction of the parameters of the dense deep neural network yet retain the same performance”.
>
> > ***How do neural pathways/sub-networks interact with different tasks?***
>
> This is related to (3), where we clarify the purpose of using mask,m and how it activates specific pathway for a specific task.
>
> How to guarantee or justify that the obtained sub-network is the right solution?
> Pruning methods hypothesis, there are many sub-network that can lead to equivalent performance. There are different methods to find sub-networks developed in pruning methods that mostly provide empirical guarantees but none of them provide a theoretical guarantee of superior performance. Similarly, we also justify the task-specific sub-network empirically through performance.
>
> > ***How are the learned skills reused/transferred among tasks?***
>
> Our proposed method does not aim to skill transfer. We understand that it is natural to expect a multitask method to reuse/transfer skills but NPF does not explicitly require/do any skill transfer. In Section 5.3, we discuss how NPF shares trainable neural-network parameters among tasks and “this
> disproves a conventional class of thinking that the success of multitask in neural networks depends on relevant data sharing (Yu et al., 2021; Fifty et al., 2021) or similar task training (Sodhani et al., 2021; Yang et al., 2020). Rather, neural networks are capable of learning multiple tasks simultaneously as long as the neurons are wired properly.”

---

### Official Review · Reviewer_6eEZ · 2022-10-24

**Confidence:** 2
**Correctness:** 2
**Technical Novelty And Significance:** 3
**Empirical Novelty And Significance:** 2
**Recommendation:** 5

**Clarity, Quality, Novelty And Reproducibility:**

In terms of originality, Pathway is an interesting idea as an overall framework for AI system to handle discrepency between tasks, domains, etc, which is first proposed by Jeff Dean to my knowledge. That been said, it is still interesting to discuss possible ways to actually implement the idea to solve problems and address difficulties, like how to find the optimal pathways for each task or domain or even sample.

In terms of concreteness, the proposed model aims at achieving similar performance using sparse sub-network in comparison with full dense network. Thus the experiment in Figure 3 shows inferior results with proposed model in comparison with baseline. It needs to further clarify that how significant it this performance difference.

**Strength And Weaknesses:**

I do have some concerns regarding this paper, especially in terms of the clarity.

The clarity of the paper needs to be improved.
1. How is global-model optimized in Algorithm 1? Providing an equation would be very helpful.
2. The most important equation (1) in paper is not clear to me. By my understanding, m_n is the mask for n_th task, while S(theta; D) is the objective function. Why these two terms are equivalent?
3. Equation 2 is also unclear to me. It envolves two techniques to stablized the training, keep a temporary replay buffer TD_n and re-confriguring using the average of the previous M iteration scores. In paper, it seems that keeping a temporary replay buffer aims to keep only the most-recent samples from agent to update the mask, but averaging between several iteration is using samples span longer history, which seems contradictory. How would you interpret it?
4. For results in Table 3, how could variance of BCQ with NPF be 0? Also, since it is normalized, I'm curious of the original performance of BCQ and IQL algorithm.



**Summary Of The Paper:**

In this paper, the authors proposed a multitask RL framework by learning multiple pathways through a single network, called Neural Pathway Framework (NPF), which aims to find a task-specific optimized mask. Started with a method from a network pruning literature, the authors further proposed to use replay buffer and average multiple iteratures to stablize training. Experimental results shows performance improvement with reduced parameter count.

**Summary Of The Review:**

The paper studies an interesting problem and gives a potential solution. My major concerns are about clarity. I might will change my recommendation after things are more clearer.

---

> ### Author Response · Authors · 2022-11-19
> **Authors' Response to Reviewer 6eEZ (Part 2/2)**
>
> **Global-model update in Algorithm 1**:
> > ***“How is global-model optimized in Algorithm 1? Providing an equation would be very helpful”***
>
> The global model is updated process asynchronously (discussed in section 4.2). We run all the processes on a single GPU and thus we are able to share the global model with all the parallel processes, which enables us to use Hogwild [1, 2] style global parameter updates. The convergence of the global parameter in a such method is proved in RL [2,3].
>
>
> **Correction on equation (1)**:
> > ***“The most important equation (1) in paper is not clear to me. By my understanding, mn is the mask for nth task, while S(theta; D) is the objective function. Why these two terms are equivalent?”***
>
> We thank the reviewer for pointing this out. The equation indeed misrepresents our method. We have added the correction in the updated version of the paper. We re-write this as "m_n = T_K(S(theta; D))", where, T_k is defined as the “top-k” operator, function identifies the top k parameter and sets the positional value as 1 and the rest 0 allowing us to have a matrix m_n.
>
>
>
>
> **Explanation on the online pathway update**:
> > *“In paper, it seems that keeping a temporary replay buffer aims to keep only the most-recent samples from agent to update the mask, but averaging between several iteration is using samples span longer history, which seems contradictory. How would you interpret it?”*
>
> The effects of configuring/updating pathways using  (1) samples collected from M previous iterations and (2) averaging M previous scores S(\theta, D) are different. In an online RL setting, we follow procedure (2) and we explain why.
>
> In online setting, we want to focus on the most recent samples collected by the updated policy under the assumption that as our policy improves, the quality of recently collected samples gets better. That's the reason why we do not want to use prior historical data in pathway discovery, especially in the initial stage of online training, where the agent takes random action to explore the environment. On the other hand, this simple score averaging stabilizes the pathway update. An average over M scores allows selection parameters that are historically proven to be more important and stabilizes the pathway update by avoiding sudden large changes.
>
> We also want to point out that intuition is similar to having an additional target critic network in the on-policy actor-critic method [4] where we slow down the target critic update to avoid an abrupt change in value space and use the most recent data for network update.
>
>
> **Explaining BCQ+NPF performance variance in MT10 and performance of original/dense BCQ algorithm**:
> > ***“For results in Table 3, how could variance of BCQ with NPF be 0? Also, since it is normalized, I'm curious of the original performance of BCQ and IQL algorithm.”***
>
> In the MetaWorld experiment, the evaluation criterion is success rate which is binary when evaluate, either it reaches the goal or does not. For MT10 tasks, 100 means BCQ reaches the goal for all the tasks all the time. The original performance of the BCQ and IQL algorithms is presented in Table -3 as  "offline MT"
>
>
>
> **Further performance comparison of different sparse techniques to explain the performance difference in Figure 3**:
>
> > *“The experiment in Figure 3 shows inferior results with proposed model in comparison with baseline. It needs to further clarify that how significant it this performance difference.”*
>
> Figure 3 compares the performance of SAC with different pruning methods (parameter counts 5%) in online RL setting, where the baseline is the final episodic return of the dense model (parameter count 100%). As can be seen in Figure (3b) and (later in Appendix Figure 14) with M=10 (averaging m 10 scores S(\theta, D)) we are able to achieve performance that is closer to dense mode. To compare our method's significance we further in normalized score w.r.t the dense model performance in Table 8.
> In terms of single-task performance, there is a scope for improvement but our focus in this paper is to use sub-networks as a way to learn multiple tasks, where we show despite the reduced number of parameters (and reduced single-task performance in online RL setting) it provides comparable performance to the baseline multitask methods in Table 4.
>
> With the improvement of the performance of the sub-network, we do expect an improvement in multitasking performance as well.
>
> We want also want to point out, this is the first paper in online RL to propose a method (DAPD) that allows getting a comparable performance with the original dense network using only 5% of the parameters.
>
>
> [1] Hogwild!: A lock-free approach to parallelizing stochastic gradient descent
> [2] Asynchronous Methods for Deep Reinforcement Learning
> [3] Asynchronous stochastic approximation and q-learning
> [4 Proximal Policy Optimization Algorithms]

---

> ### Author Response · Authors · 2022-11-19
> **Authors' Response to Reviewer 6eEZ (part 1/2)**
>
> We want to thank the reviewer for the thorough review and asking the most important questions allowing us to explain further our methods and making the necessary corrections.
>
>
> **Global-model update in Algorithm 1**:
> > ***“How is global-model optimized in Algorithm 1? Providing an equation would be very helpful”***
>
> The global model is updated process asynchronously (discussed in section 4.2). We run all the processes on a single GPU and thus we are able to share the global model with all the parallel processes, which enables us to use Hogwild [1, 2] style global parameter updates. The convergence of the global parameter in a such method is proved in RL [2,3].
>
> [1] Hogwild!: A lock-free approach to parallelizing stochastic gradient descent
> [2] Asynchronous Methods for Deep Reinforcement Learning
> [3] Asynchronous stochastic approximation and q-learning
> [4 Proximal Policy Optimization Algorithms]
>
> **Correction on equation (1)**:
> > ***“The most important equation (1) in paper is not clear to me. By my understanding, mn is the mask for nth task, while S(theta; D) is the objective function. Why these two terms are equivalent?”***
>
> We thank the reviewer for pointing this out. The equation indeed misrepresents our method. We have added the correction in the updated version of the paper. We re-write this as "m_n = T_K(S(theta; D))", where, T_k is defined as the “top-k” operator, function identifies the top k parameter and sets the positional value as 1 and the rest 0 allowing us to have a matrix m_n.
>
>
>
>
> **Explanation on the online pathway update**:
> > *“In paper, it seems that keeping a temporary replay buffer aims to keep only the most-recent samples from agent to update the mask, but averaging between several iteration is using samples span longer history, which seems contradictory. How would you interpret it?”*
>
> The effects of configuring/updating pathways using  (1) samples collected from M previous iterations and (2) averaging M previous scores S(\theta, D) are different. In an online RL setting, we follow procedure (2) and we explain why.
>
> In online setting, we want to focus on the most recent samples collected by the updated policy under the assumption that as our policy improves, the quality of recently collected samples gets better. That's the reason why we do not want to use prior historical data in pathway discovery, especially in the initial stage of online training, where the agent takes random action to explore the environment. On the other hand, this simple score averaging stabilizes the pathway update. An average over M scores allows selection parameters that are historically proven to be more important and stabilizes the pathway update by avoiding sudden large changes.
>
> We also want to point out that intuition is similar to having an additional target critic network in the on-policy actor-critic method [4] where we slow down the target critic update to avoid an abrupt change in value space and use the most recent data for network update.
>
> **Explaining BCQ+NPF performance variance in MT10 and performance of original/dense BCQ algorithm**:
> > ***“For results in Table 3, how could variance of BCQ with NPF be 0? Also, since it is normalized, I'm curious of the original performance of BCQ and IQL algorithm.”***
>
> In the MetaWorld experiment, the evaluation criterion is success rate which is binary when evaluate, either it reaches the goal or does not. For MT10 tasks, 100 means BCQ reaches the goal for all the tasks all the time. The original performance of the BCQ and IQL algorithms is presented in Table -3 as  "offline MT"

---

### Official Review · Reviewer_jX9C · 2022-10-24

**Confidence:** 4
**Correctness:** 3
**Technical Novelty And Significance:** 3
**Empirical Novelty And Significance:** 4
**Recommendation:** 6

**Clarity, Quality, Novelty And Reproducibility:**

The paper is mostly well written and mostly correct (see comments below).
The evaluation is mostly solid and the appendix contains enough information to reproduce the experiments.

**Minor comments:**
- sec.3: "naivelt"
- sec.3: say explicitly whether sharing the same transition is a necessary assumption
- sec.4.1: $\frac{\partial\mathcal L}{\partial \theta_q}$ appears in different equations with different meanings. Clarify this by using $\frac{\partial\mathcal L(\theta_0)}{\partial \theta_q}$ and/or $\frac{\partial\mathcal L(\theta_0; D_n)}{\partial \theta_q}$
- eq.1: $m_n = S(...)$ appears incorrect. It should be something like $m_n = \text{argmax}_{q} S(...)$.
- sec.4.2: shortly explain what "synchronous parameter updates" are
- sec.4.2: "function 1" -> "Equation (1)"
- alg.1: clarify where the $n$ in the training loop comes from (e.g. run all tasks $n$ in parallel) and when exactly the optimizer is executed (after all gradients have been summed). Currently the parallelization part is more confusing than clarifying. Adding an inner loop might help.
- eq.2: it is completely unclear what the sum-variable $m$ refers to here, as it does not appear in the equation and overloads the mask $m$. Better define the sum over variable $k$ and (a) use $TD_n^{t-M:t}$ to denote that you are referring to replay buffer length or (b) define some $s_n^t$ as score at time $t$ and then select the mask based on the average last M scores $\frac{1}{M} \sum_{k=0}^{M-1} s_n^{t-k}$ or (c) define an online update $\bar s_n^t = (1-\alpha) \bar s_n^{{t-1}} + \alpha s_n^t$.
- alg.2: the algorithm does not reveal how "the last M iterations" affect the masks and does not show how $D_n$ and $TD_n$ differ (they both just get all the data, what is the difference?).
- tab.2: there need to be other baselines (like in tab.3) for comparison of the Mujoco experiments.
- fig.5: the caption almost touches the main text
- p. 6: the paragraph "Online Multitask Training and Addressing Gradient Interference" is in a weird place. It would make more sense in the beginning of Section 4.
- sec.5.1: the text says "the mean performance with 95% confidence interval", but the caption fo Table 3 says "mean and std over 10 seeds". Which is it?
- sec.5.1: the MT baseline is ambiguous: do you train one network for all tasks (which should not work without a task indicator) or one network per task?
- fig.6: figure almost touches the main text

**Strength And Weaknesses:**

**Strength**

The paper presents an innovative approach to learn and represent multiple tasks in one network. The approach is simple and can be adapted into many algorithms. Results look generally good.

**Weaknesses**

The idea is, to the best of my knowledge, novel, but not very deep. Following the lottery hypothesis, the authors just select parameter-masks and keep them (in the offline setting) during training fixed. In the online setting the authors identify that the change in loss requires them to recompute the mask based on newer data. However, the implementation (averaging the last $M$ scores, stopping criterion) sounds very simplistic and there is no analysis on the dynamics of selected masks, ways to deal with this non-stationarity and whether the mask eventually converges, just as the policy does. More analysis on this would strengthen the paper significantly. Furthermore, there are some questions I would like to have answered:
1. You emphasize that the training data changes the loss a lot, but in RL the loss is also changed e.g. by bootstrapping: at time $t$ the loss depends on the value function(s) with parameters $\theta_t$. Have you evaluated this? How is the effect of this on the mask selection?
2. What is the "continual pathway configuration" (referred to later as "parallel learning") in Figure 4? Is it NPF without the pretraining stage or training without mask or something else?
3. How significant are the results? It looks as if the standard deviations overlap a lot. Can you make a statistical test whether your method is significantly better (p <= 0.05) than the next best baseline?

**Summary Of The Paper:**

The paper aims to learn multiple RL tasks with the same network. SNIP is used to prune all but the 5% most "relevant" parameters of the network, yielding a mask for every task. Tasks can share parameters, and therefore all tasks have to be trained simultaneously. The authors show that SNIP depends strongly on the data set, and propose to regularly change the task-masks based on the newest data for online RL scenarios. The new method NPF is evaluated on 3 Mujoco and the Meta-World multi-task benchmarks with the BCQ, IQL and SAC algorithms and compared with other baselines. Results show that NPF is competitive with other multi-task approaches.


**Summary Of The Review:**

I really liked this paper. It is not totally clear whether the method will allow something that would otherwise be very hard in multi-task RL, but it is a nice idea and in my opinion well evaluated. The only weak point is a relatively low novelty and a lack of theoretical analysis. However, unless another reviewer finds something really wrong with it, I recommend to accept the paper.

---

> ### Author Response · Authors · 2022-11-19
> **Authors' Response to Reviewer jX9C (Part 2/2)**
>
>
> > ***“You emphasize that the training data changes the loss a lot, but in RL the loss is also changed e.g. by bootstrapping: at time t the loss depends on the value function(s) with parameters θt .Have you evaluated this? How is the effect of this on the mask selection?”***
>
>
> We select mask/pathway for policy and value network independently, where each network has its own objective function. The SNIP criterion takes into account the training dataset and the loss to calculate a score for the saliency of each parameter in reducing the loss. Thus, the effects of bootstrapping is taken into account with the snip criterion. We do however agree with the reviewer that disentangling the effects of bootstrapping will be an interesting study, which we feel is currently beyond the scope of our work.
> However, if the reviewer feels that his query wasn’t fully addressed, we ask the reviewer to please further clarify the question and we would try to provide a clearer response.
>
>
> > ***What is the "continual pathway configuration" (referred to later as "parallel learning") in Figure 4? Is it NPF without the pretraining stage or training without mask or something else?:***
>
> We added further context to clarify the difference between the two methods. In the MetaWorld experiment, we find some of the harder tasks never really learn due to gradient interference and to overcome gradient interference, we run a pretraining loop to learn pathways which we denote as “pre-configured pathways” that we keep fixed for the rest of the training. On the other hand, for regular pathway training, we continue to update the pathway throughout the network training and we refer to it as “continual pathway configuration”
>
> > ***How significant are the results? It looks as if the standard deviations overlap a lot. Can you make a statistical test whether your method is significantly better (p <= 0.05) than the next best baseline?***
>
> In the following Table we show the performance comparison on MT10 task from MetaWorld. The performance with baseline is compared in Table 4. We do have standard deviationsget but as comapred in Table 4 it also allow us to higher FLOP rates and requires significantly small amount parameter during evaluation.
> | Experiment | Continual pathway configuration| Pre-configured pathway |
> | -------- | -------- | -------- |
> | MT10     | 59.5 $\pm$  6.7   | 77.4 $\pm$ 13.12     |
>
>
> > ***sec.5.1: the MT baseline is ambiguous: do you train one network for all tasks (which should not work without a task indicator) or one network per task?***
>
> In MT baseline, we train single network for all the tasks. Since we are using only expert data for this experiment and training offline, this works suprising well. We further added performance comparison with mixed dataset on the updated version of the paper (Appendix C.2)

---

> > ### Comment · Reviewer_jX9C · 2022-11-27
> > **Thanks for the rebuttal**
> >
> > Thanks to the authors for their extensive answers. It seems that the main paper has not been changed and the claimed changes in the Appendix are not visually identifiable. In future submissions, please mark changes in the rebuttal with another color.
> >
> > I can follow most arguments, except two:
> >
> > >  The SNIP criterion takes into account the training dataset and the loss to calculate a score for the saliency of each parameter in reducing the loss. Thus, the effects of bootstrapping is taken into account with the snip criterion.
> >
> > If I understand you correctly, this was not what I was talking about: the problem of value regression is that at each transition the "label" contains the value of the next time-step. When the value changes throughout training (which also happens in offline-learning), so does the loss and therefore the SNIP mask (for the same initial parameters). While your paper identifies that online-RL leads to a non-stationary transition-distribution, even in offline-RL the agent has to deal with a non-stationary label-distribution. This *should* have an affect on NPF, which should be discussed and evaluated. I also wonder whether a SNIP mask based on the initial parameters is truly the best choice when the parameters have already changed significantly.
> >
> > > We see for M=10, the reconfiguration of the pathway stops earlier than the other methods. Following our algorithm-2, it means, for M=10 the agent has reached/converged the threshold performance earlier than others where we stop updating the pathway. This indicates a stable pathway update through averaging scores indeed leads to "stable performance".
> >
> > While I appreciate the additional experiment, it does not fix the theoretical issues of the paper as is. First, distance from initial mask is not the best measure to show convergence. Second, the observation M10 does not visibly "stop earlier". The large standard deviation does not allow such a claim and seems to widen at the end. Lastly, the fact that you saw it necessary to include a stopping criterion in Algorithm 2 gives readers the impression that the masks do in fact *not* converge.

---

> > > ### Author Response · Authors · 2022-12-01
> > > **Authors' Response to Reviewer jX9C**
> > >
> > > We thank the reviewer for further clarifying his concern.
> > >
> > > ---
> > >
> > > > ***While your paper identifies that online-RL leads to a non-stationary transition-distribution, even in offline-RL the agent has to deal with a non-stationary label-distribution. This should have an affect on NPF, which should be discussed and evaluated. I also wonder whether a SNIP mask based on the initial parameters is truly the best choice when the parameters have already changed significantly.***
> > >
> > >
> > > * Under ***fixed data-distribution***, there are pruning methods that follow an iterative optimization procedure[1,2] which if used in offline RL, can explicitly deal with the non-stationary label distribution.
> > > * But in this setting, single shot pruning (subnetwork/mask discovery based on the initial parameter) such as SNIP[3] is found to be equally effective whereas reduces the extra compute.
> > >
> > >
> > > > ***distance from initial mask is not the best measure to show convergence.***
> > > * We agree with the reviewer but the purpose of the provided figure is rather to show the steady change (for M=10) in mask configuration which allows the agent to reach the threshold performance in fewer gradient updates (table below).
> > >
> > > | Environment |    M=1    | M=5      | M=10     |
> > > | --------    | -------- | -------- | --------  |
> > > | HalfCheetah Forward  | 141 $\pm$ 31k    | 108 $\pm$ 11k    |  **100 $\pm$ 20k** |
> > >
> > >
> > >
> > > > ***Lastly, the fact that you saw it necessary to include a stopping criterion in Algorithm 2 gives readers the impression that the masks do in fact not converge.***
> > > * Pruning literature hypothesizes [4] existence of a ***set of lottery subnetworks*** (where each subnetwork is expected to perform the same as the dense network) and as we use the snip criterion in online RL, it does not find scores that converge to a unique mask. We conjecture that in online RL, without the stopping criteria, it ends up switching among these subnetworks and that leads to high variance in performance [figure 3(a) DAPD (green)]. The stopping criteria allow us to select one of these subnetworks and optimize corresponding parameters.
> > > * We would like to point out, having heuristic scheduling as a stopping criterion is commonly used in the iterative pruning literature [1,2].
> > >     * In the pruning literature the state-of-the-art iterative pruning methods rely on multiple computationally costly train-prune-finetune cycles where a 100% dense network is ***gradually*** pruned following a heuristic scheduling. At each iteration, a percentage of the network is pruned and finetuned to finally generate a sub-network. Moreover, a range of pruning heuristics is widely used, such as learning rate and weight rewinding which rewinds the weights of the networks to a certain epoch early in training to arrive at a better network.
> > >     * Our heuristic to have a stopping criterion takes inspiration from iterative pruning literature and is a natural extension of the SNIP criterion to deal with changing data distribution.
> > >
> > > [1] Linear Mode Connectivity and the Lottery Ticket Hypothesis
> > > [2] Comparing Rewinding and Fine-tuning in Neural Network Pruning
> > > [3] SNIP: Single-shot Network Pruning based on Connection Sensitivity
> > > [4] The Lottery Ticket Hypothesis: Finding Sparse, Trainable Neural Networks
> > >
> > >
> > > **Updated version of the paper**: Unfortunately, due to an internet error, we could not upload the updated version of the draft before the AOE deadline. Once this came to our attention, we sent the updated draft to the programme committee (17 mins past the deadline) and later to Area-chair but have not received any response yet. Here is the link for the revised paper: https://drive.google.com/file/d/1f88LJVXncBoubP0Yoirclpy0OPr5QD7s/view?usp=sharing

---

> ### Author Response · Authors · 2022-11-19
> **Authors' Response to Reviewer jX9C (Part 1/2)**
>
> We sincerely thank the reviewer for their time in reviewing the paper and providing in-depth constructive feedback. We are elated that the reviewer finds our method innovative and recognizes the simplicity and adaptability of the algorithm.
>
> In an Online setting, to achieve a stable performance we propose DAPD where we average the parameter importance (using equation 2) over multiple iterations (M). The reviewer expects further analysis “on the dynamics of selected masks, ways to deal with this non-stationarity” and raises concerns about the “mask convergence”.
>
> We agree with the reviewer that “averaging” is a simple solution. But we find it very effective and computationally efficient. To address reviewers' concerns we do further analysis on (1) how this pathway/mask changes over time and (2) how it affects the performance.
>
> We run an experiment (shown in figure below) to see the pathway (parameter fo the subnetwork) change from its initialization over time and it indicates a very drastic change as the policy and its corresponding data distribution changes. Thus it requires a more stable pathway update.
>
> Figure shows the percentage of change in the pathway parameters from the random initialization. We learn two things from this experiment.
> 1. The change in a pathway is slower as M increases, which leads to a more “stable pathway update”. An average over M scores allows selecting parameters that are historically proven to be more important and avoids sudden large change.
> 2. We see for M=10, the reconfiguration of the pathway stops earlier than the other methods. Following our algorithm-2, it means, for M=10 the agent has reached/converged the threshold performance earlier than others where we stop updating the pathway. This indicates a stable pathway update through averaging scores indeed leads to “stable performance”.
>
> ![](https://i.imgur.com/z7hmM70.png =300x200)
>
> In table below we provide the number of steps required for Halfcheetah-Forward task to reach 2k episodic return.
>
>
> | Environment |    M=1    | M=5      | M=10     |
> | --------    | -------- | -------- | --------  |
> | HalfCheetah Forward  | 141 $\pm$ 31k    | 108 $\pm$ 11k    |  **100 $\pm$ 20k** |
>
>
>
> We also want to point out that intuition is similar to having an additional target critic network in the actor-critic method to avoid drastic changes in value space. Since this is the first paper, we believe it is a good exploratory first step and will further inspire more methods to explore better ideas.
>
>
> **Proof of pathway/mask convergence:**
>
> * Pruning methods hypothesis [1], there are many sub-network that can lead to equivalent performance. There are different methods to find sub-networks developed in pruning methods mostly provide empirical guarantees through performance but none of them provide a theoretical guarantee of superior performance
> * We also empirically found this simple method to be effective and the learning curve [figure 3(b), 14] in our experiment suggests, the found pathway leads to a stable performance.
>
> ---
> [1] The Lottery Ticket Hypothesis: Finding Sparse, Trainable Neural Networks

---

### Official Review · Reviewer_Q8xY · 2022-10-25

**Confidence:** 3
**Correctness:** 2
**Technical Novelty And Significance:** 2
**Empirical Novelty And Significance:** 2
**Recommendation:** 3

**Clarity, Quality, Novelty And Reproducibility:**

The Clarity should be further improved. This paper has a lot of grammar issues, such as 'nonstationaritym',  'radient interference', and 'of group the parameters'.

The idea is composed of several existing techniques, which makes it somewhat not novel. The pruning technique used in this paper does not consider the difference from the single-task setting.

Open-sourced code is provided.

**Strength And Weaknesses:**

Strengths:

The idea of introducing sparse training to multi-task RL is a promising direction.

The proposed framework is combined with both online and offline RL methods, showing the generality.

Weaknesses:

The literature review is not extensive, missing the discussion and comparison of related works [1-3].

1) Some technical details are not convincing: not considering Data distributional shift in offline RL setting. Data distributional shift is a common problem in offline RL since the data set is collected by some unknown policy. When the trained policy interacts with the environment, there exist some states/actions out of distribution where the policy may behave badly.

2) The masks are fixed during training. However, it's more reasonable to adjust the mask since the sensitive measure is not static over the whole state space. Also, it is not static during different training stages. On contrary, if the mask is adaptive changing, it may cause unstable training.

3) The experimental results should focus more on the benefits of sparse training regarding the training time/resources saved. For example, the parameter reduction results in offline RL are not provided.

Experiments should consider the full MT50 task set and compare the proposed method with CAgrad [4].

[1] Sparse Multi-Task Reinforcement Learning.
[2] Multi-task Batch Reinforcement Learning with Metric Learning.
[3] Hub-Pathway: Transfer Learning from A Hub of Pre-trained Models.
[4] Conflict-averse gradient descent for multi-task learning.


**Summary Of The Paper:**

This paper proposes to use masks in multi-task training so that only a part of network parameters are used for a specific task. The framework considers combining both online and offline RL methods. Experiments show that the proposed method significantly reduces the training parameters and achieves competitive performance compared with previous methods.

**Summary Of The Review:**

Overall, this paper points out a promising direction in MTRL and provides relatively extensive experimental results. However, as listed in the pros and cons part, the novelty, technical issues, and missing related works make this paper lean to reject.

---

> ### Author Response · Authors · 2022-11-19
> **Authors' response to reviewer Q8xY (Part 2/2)**
>
>
> The reviewer also advised mentioning other related work [2-4].  We argue that [2-4] are literature from MetaRL that only works under the assumption that all the tasks are very similar in nature.
>
> > "The idea is composed of several existing techniques, which makes it somewhat not novel"
>
> * We want to point out pruned/sparse neural network applied in multitask setting in novel to our paper and we show its significance in both offline and online settings.
>
> * We do agree that our method leverages existing techniques. The idea of SNIP itself is based on a very old saliency criterion [6]. However, the idea of utilizing a saliency criterion in the multitusk setting for finding task-specific subnetworks is a novel one. This also opens the door to further research in multi-task learning leveraging other saliency criteria.
>
> * Our method works in both online and offline RL settings without changing the learning objective of the algorithms. Unlike conventional approaches, where multitask methods mostly rely on task similarity, modular network training, restoring into complicated gradient update, or underlying context learning, we propose a new way to tackle multitask problems and show function approximation with a single deep neural network is sufficient.
>
> * We empirically show that the single-shot lottery network search fails under data distribution shift We overcome this by proposing data adaptive pathway discovery.
>
> Hence, we believe the contributions are novel and useful.
>
>
> > ***“The pruning technique used in this paper does not consider the difference from the single-task setting”***
>
> That is one of the benefits we achieved using our proposed method.  Using NFP, we do not have to consider additional objectives as we move to multitask training. We consider finding pathways (a subset of parameters) for each task as an independent process and empirically show this can be easily extended into multitasking learning through targeted gradient updates using the masks.
>
>
> [1] Conflict-averse gradient descent for multi-task learning.
> [2] Sparse Multi-Task Reinforcement Learning.
> [3] Multi-task Batch Reinforcement Learning with Metric Learning.
> [4] Hub-Pathway: Transfer Learning from A Hub of Pre-trained Models.
> [5] Multi-Task Reinforcement Learning with Context-based Representations
> [6] Skeletonization: A Technique for Trimming the Fat from a Network via Relevance Assessment.
> [7] https://github.com/facebookresearch/fvcore

---

> ### Author Response · Authors · 2022-11-19
> **Authors' response to reviewer Q8xY (Part 1/2)**
>
> We appreciate that the reviewer finds the experimental results extensive and the sparse training in multitask RL a promising direction.
>
> We thank the reviewer for thoughtful suggestions on the literature review and for asking for further clarification on our method, especially since the reviewer raises the concern about 3 things (1) what happens in data distribution shift in offline RL, (2) how the mask is updated during training and "if the mask is adaptive changing", whether or not it may cause unstable training, (3) what kind of gain we achieve from sparse training compared to baseline.
>
> **Data distribution shift in offline RL and mask training:**
> We agree with the reviewer that there can be data distribution shift in offline data and most of the offline methods tries to guarantee the best performance under such circumstances. But data distribution shift in "offline RL" setting does not affect discovering neural pathways as it does in “online RL”. Here we explain why.
>
> In an offline setting, the "data distribution is fixed" and known prior (no further changes in data distribution during the training) and thus we can discover the neural pathway beforehand at a single shot using equation 1. That is, in the Offline RL setting we already know the dataset we want to learn the behaviour from. Hence, the saliency criterion we use is able to generate the appropriate pathway.
>
> On the other hand in online training the "data distribution keeps changing during training" and better policy leads to better data distribution, thus it is important to update the pathway based on the data coming from the updated policy. Further discussed in Section 4.3
>
> To support our claim, we run further experiments (added in the updated version of the paper (Appendix C.2)) in offline RL multitask under different data distribution shifts and compare the performance with the baseline. We use mixed and imperfect dataset where in (i) medium: dataset collected from suboptimal agent trained for 300k gradient steps, (ii) medium-expert:  mixing equal amounts of expert demonstrations and suboptimal data and (iii) expert-replay: recording all the sample observed by the agent during training and represents a dataset generating from a mixture of many distributions. To handle this mixture of distrition we take a larger batch of sample (x10) to evaluate $S(\theta,D)$.
>
> ![](https://i.imgur.com/CU8AI4Y.png =400x200)
>
> **Benefit of spare training and parameter reduction in offline RL:**
> To address the reviewer's concern we add the parameter reduction compared to the baseline in offline RL experiment in Table 2. To show the benefit of sparse training,  We show that the NPF requires the lowest floating point operation (FLOP) counts when compared to offline (Table 3) and online (Table 4) baseline. A lower FLOP count is linearly proportional to faster inference. For any given task we use [7] to compute FLOP for each method.
>
> **Literature review:**
> We believe we have discussed the most recent relevant multitask RL papers in the literature review. We thank the reviewer for pointing out [1] and suggesting further comparison. We could not compare MT50 due to computation cost (5 days compute time) but we compare the MT10 performance with the baseline reported in [5] and for a fair comparison we do this experiment on the same MetaWorld environment as the baseline [5] (https://github.com/facebookresearch/mtrl, commit “af8417bfc82a3e249b4b02156518d775f29eb289”). The results are reported in Table 4 and can also be downloaded from the following link:
> https://drive.google.com/file/d/1K6XlccOuU3TaeALcnzR93qbBZ-kRRdOM/view?usp=share_link

---

### Decision · Program_Chairs · 2023-01-20

**Decision:**

Reject

**Justification For Why Not Higher Score:**

The work has many deficiencies, failing to provide a clear value.

**Justification For Why Not Lower Score:**

N/A

**Metareview: Summary, Strengths And Weaknesses:**


This paper addresses the problem of multi-task learning using a single network where each task uses a subnetwork much smaller than they usually use. The authors use an existing method for calculating connection sensitivity, which is utilized to create separate masks for the tasks. I appreciate that the authors aim to address an important problem. However, the results and the presentation did not create a clear value. The message and many details of the work did not get across well. Moreover, a clear intuition needs to be included on why an initial computation of mask worked well and better than continuing to change the mask. I strongly encourage the authors to work on the exposition of the idea clearly and justify the steps of the method more carefully and rigorously for future submission.